# Applications of Sustainable Hybrid Energy Harvesting: A Review

**Hamna Shaukat [1], Ahsan Ali [2], Shaukat Ali [2], Wael A. Altabey [3,4,*], Mohammad Noori [5,6,*] and Sallam A. Kouritem [3]**

1 Department of Chemical and Energy Engineering, Pak-Austria Fachhochschule: Institute of Applied Sciences and Technology, Mang, Haripur 22621, Pakistan; b20f0001che001@fcm3.paf-iast.edu.pk
2 Department of Mechatronics Engineering, University of Wah, Wah Cantonment 47040, Pakistan; ahsan.ali@wecuw.edu.pk (A.A.); shaukat.ali@wecuw.edu.pk (S.A.)
3 Department of Mechanical Engineering, Faculty of Engineering, Alexandria University, Alexandria 21544, Egypt; sallam.kouritem@alexu.edu.eg
4 International Institute of Urban Systems Engineering (IIUSE), Southeast University, Nanjing 210096, China
5 Department of Mechanical Engineering, California Polytechnic State University, San Luis Obispo, CA 93405, USA
6 School of Civil Engineering, University of Leeds, Leeds LS2 9JT, UK
* Correspondence: wael.altabey@gmail.com (W.A.A.); mnoori@calpoly.edu (M.N.); Tel.: +86-173-6847-6644 (W.A.A.); +18-0590-32411 (M.N.)

**Abstract:** This paper provides a short review of sustainable hybrid energy harvesting and its applications. The potential usage of self-powered wireless sensor (WSN) systems has recently drawn a lot of attention to sustainable energy harvesting. The objective of this research is to determine the potential of hybrid energy harvesters to help single energy harvesters overcome their energy deficiency problems. The major findings of the study demonstrate how hybrid energy harvesting, which integrates various energy conversion technologies, may increase power outputs, and improve space utilization efficiency. Hybrid energy harvesting involves collecting energy from multiple sources and converting it into electrical energy using various transduction mechanisms. By properly integrating different energy conversion technologies, hybridization can significantly increase power outputs and improve space utilization efficiency. Here, we present a review of recent progress in hybrid energy-harvesting systems for sustainable green energy harvesting and their applications in different fields. This paper starts with an introduction to hybrid energy harvesting, showing different hybrid energy harvester configurations, i.e., the integration of piezoelectric and electromagnetic energy harvesters; the integration of piezoelectric and triboelectric energy harvesters; the integration of piezoelectric, triboelectric, and electromagnetic energy harvesters; and others. The output performance of common hybrid systems that are reported in the literature is also outlined in this review. Afterwards, various potential applications of hybrid energy harvesting are discussed, showing the practical attainability of the technology. Finally, this paper concludes by making recommendations for future research to overcome the difficulties in developing hybrid energy harvesters. The recommendations revolve around improving energy conversion efficiency, developing advanced integration techniques, and investigating new hybrid configurations. Overall, this study offers insightful information on sustainable hybrid energy harvesting together with quantitative information, numerical findings, and useful research recommendations that progress and promote the use of this technology.

**Keywords:** energy harvesting; hybrid energy harvesters; sustainable energy harvesting; energy conversion; green energy harvesting

## 1. Introduction

Using sustainable/renewable energy sources, such as light, wind, heat, waves, rotation, or vibration, becomes a viable and effective solution to the world's energy crisis [1–4].

Environmental energy-harvesting methods have been the subject of numerous studies and efforts over the past years. These energy sources provide electricity for household and industrial use, thus addressing local power shortages. Because of the challenge of connecting electrical cables with various sensors, powering wireless sensor node systems is a concern [5–7]. However, a source of power without the self-powering ability limits the sustainable functioning of wireless sensor systems. It creates difficulties for the users due to the low capacity of batteries [8–10]. Although the operating times of wireless sensor systems can be increased by using an ultralow power system and a high-capacity battery, they cannot ensure the system will run without interruption for a long period of time [11]. Thus, one of the innovative aspects proposed for a sustainable future society is an energy-harvesting system which transforms waste environmental energy into electrical energy [12–14]. Such energy harvesters provide sustainable power solutions by collecting ambient sustainable/renewable energy sources and converting them into electrical energy using a variety of transduction mechanisms [10,15–17], including thermoelectric, photovoltaic, pyroelectric, electromagnetic, piezoelectric, triboelectric, and other mechanisms. The development of self-charging electronics and self-powering wireless sensor systems has gained considerable interest in research on sustainable and renewable energy harvesters [18–21]. Wang et al. [22] introduced an innovative self-sustained wireless sensor network through the combination of a hybrid piezoelectric generator (PEG) and triboelectric nanogenerator (TENG). The PEG, comprising a hinged–hinged PZT bimorph and two T-shaped proof masses, generates an output power of 6.5 mW when excited at 25 Hz with 1.0 g acceleration. This power is used to light up 30 serial LEDs in sine vibration and 20 serial LEDs in shock vibration, serving as alarms for vibration and drop monitoring. Additionally, a triboelectric accelerometer demonstrates excellent linearity with a sensitivity of 15 V/g within the range of 0–1.5 g with an optimized gap of 1.5 mm. Ensuring a self-sustained power supply is a crucial objective for various applications.

There are always both artificial and natural energies present, including wind, solar, wave, machine vibration, heat, and automobile noise energies. As a result, solar, thermal, and mechanical energy-harvesting devices can coexist and continuously produce energy, as illustrated in Figure 1 [23]. To date, energy harvesters have typically been made to utilize a single source of energy. For example, photovoltaic harvesters were created to harvest light energy; to harvest thermal gradients, pyroelectric and thermoelectric harvesters were specifically created; and for harvesting kinetic energy, piezoelectric, triboelectric, electrostatic, and electromagnetic harvesters are especially helpful. A single energy harvester cannot always meet the power needs of electronic devices because its energy generation depends on the accessibility of the energy source. The environment is filled with kinetic energy from human activities, wind flows, structural and machine vibration, water waves, etc. Yet, because humans require rest, wind or water waves might not always be always present, and a machine might not run continuously, kinetic energy might be insufficient and fluctuating. Kinetic energy harvesters will not work in these situations. Thermal energy harvesters would likewise experience similar circumstances when dealing with unpredictably changing temperature gradients. Hybrid energy harvesting is gaining popularity as a response to the problem of energy deficiency among single energy harvesters. In general, it refers to both the collection of energy from various sources and the conversion of that energy into electrical energy via various transduction mechanisms. The two types of hybrid energy harvesters (HEHs) are multi-source hybrid energy harvesters and single-source hybrid energy harvesters. The result is the development of integrated multi-source HEHs with a variety of configurations and energy conversion materials [24,25]. When various power sources are accessible alternately or simultaneously, power outputs can be greatly increased. The main research focus is still on developing and creating highly efficient single-source HEHs with different energy transduction mechanisms. Emerging energy-harvesting concepts, including flexoelectric [26], flexible organic ionic diodes [27], ferro-electrets [28], mechano-radical [29], electrochemical [30], and mechanisms based on biomaterials [31], have been published recently. It has been discovered via the comparative

analysis of the energy harvesters that various harvesting mechanisms and materials may be appropriate for various application scenarios and structural configurations. In recent years, numerous researchers have provided innovative and efficient strategies for HEHs from single-source and multi-source harvesters [32]. As there are fewer publications compared to single-source harvesters, these studies still have not been thoroughly examined and summarized.

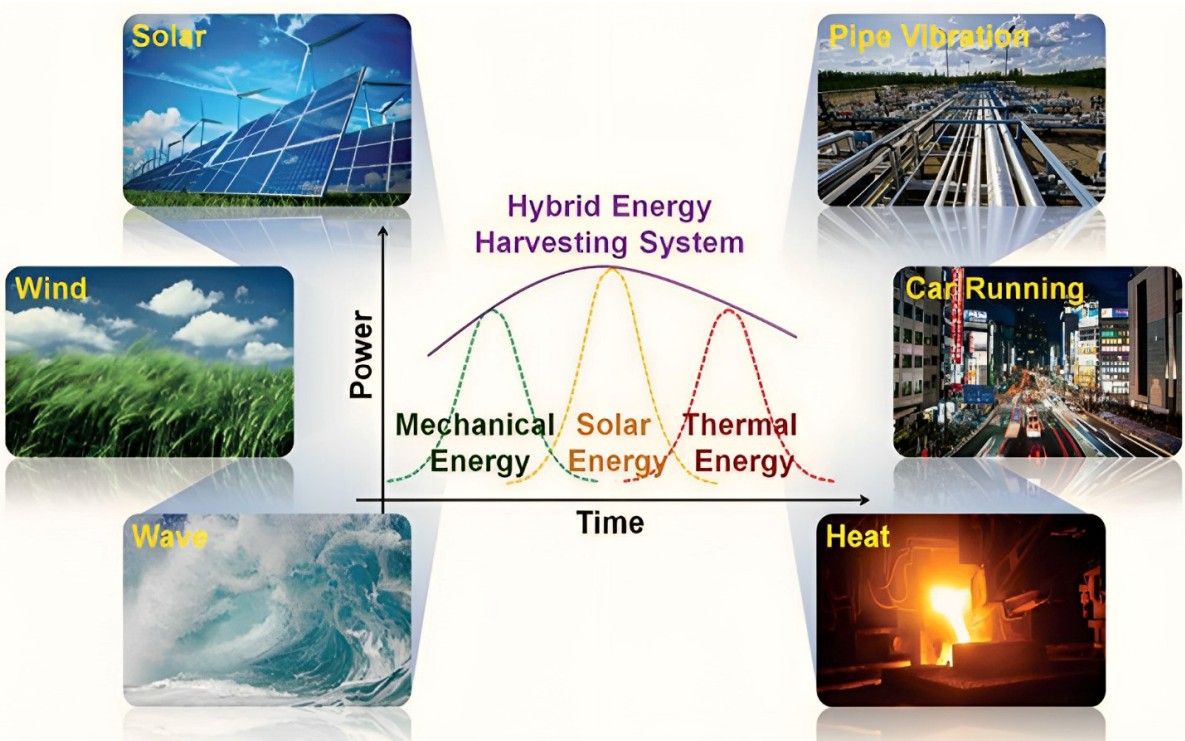

**Figure 1.** Diagram of a hybrid energy harvesting that uses both artificial and natural energies [23].

Research on hybrid energy harvesters has experienced a tremendous increase in recent years, which has resulted in significant developments in the field. A critical analysis of the existing literature identifies several trends and patterns that help to explain hybrid energy harvesting and its uses. To increase power output and boost energy efficiency, numerous studies have concentrated on investigating various combinations of sources of energy and transduction mechanisms. For instance, studies have demonstrated the advantages of integrating electromagnetic and piezoelectric energy harvesters along with the synergistic effects attained by combining electromagnetic, triboelectric, and piezoelectric energy harvesters. Comparing these hybrid systems to single-source energy harvesters, they have shown to be more capable of producing power. In addition, a deeper examination of the literature reveals the gaps and significant areas for hybrid energy harvesting. Fewer studies have examined the real-world applications and execution of hybrid energy-harvesting systems. At the same time, some have mainly concentrated on the technical aspects of the conversion of energy and power generation. It is essential to consider the requirements and limits of various application areas if one wants to grasp the advantages of hybrid energy harvesting completely. For instance, the incorporation of hybrid energy harvesters in the infrastructure and vehicles can help create self-sufficient and sustainable transportation systems in smart mobility. In the same way, in healthcare, the development of devices that are driven by hybrid energy harvesters can offer ongoing, unnoticeable health monitoring.

Additionally, a comparison of the existing studies demonstrated variability in the efficiency, reliability, and performance of various hybrid energy-harvesting systems. These differences result from the use of differed materials, configurations, design, optimization techniques and energy management approaches. The elements that lead to greater per-

formance and efficiency can be easily identified by critically analyzing and comparing these studies, thereby paving the path for further advances in hybrid energy-harvesting systems. Although hybrid energy-harvesting research has advanced significantly, there are still possibilities to expand the understanding of its uses and fill the existing gaps. Yet, this cutting-edge technology still has several significant gaps. First, maintaining compatibility and seamless integration between the various energy sources is a challenging task that calls for advanced power control and management systems. Furthermore, combining various energy-harvesting technologies into a single system may result in an increase in overall size and weight, creating problems for applications where portability and compactness are essential. In addition, as numerous conversions and storage of energy components must be included, deploying hybrid energy-harvesting systems might be costly [33–35]. To fully grasp the potential of hybrid energy harvesting and advance sustainable and effective energy solutions in the future, it is essential to overcome these challenges. Regions with high potential can be easily determined and permit the development of more practical and efficient hybrid energy-harvesting systems by conducting a more thorough analysis of the literature, examining the patterns, developments, ideas, and interactions among studies, and comparing the energy efficiencies of various approaches [36–41].

This article presents a review from the literature on the recent progress in hybrid energy harvesting and its applications, which includes smart transportation, infrastructure health monitoring, marine monitoring and development systems, aerospace engineering, healthcare monitoring, industry condition monitoring, and water purification. While already written review papers have primarily focused on single application areas, this paper offers a comprehensive performance comparison and explores the diverse applications of hybrid energy harvesting across multiple domains. In the last section, there are also some recommendations to fill the gaps and enhance hybrid energy harvesting. This review provides valuable insights to the researcher and reader on the potential and applications of the hybrid energy-harvesting systems, as well as how it helps to solve the energy-deficiency problems by providing sustainable energy-harvesting solutions, and it also explores some opportunities for future research and advancement in this area.

## 2. Hybrid Energy Harvesting

Hybrid energy harvesting integrates multiple energy conversion mechanisms into one design. The hybrid energy harvesters discussed in this paper show the integration of different harvester types to achieve synergistic effects that enhance the overall energy output. They are not just a simple combination but are designed for optimal interaction between the integrated components. The integration aims to capitalize on coupling effects during operation, thereby significantly boosting the energy output. It is essential to carefully design these systems to remove any negative impacts or interference that might potentially reduce the overall energy-harvesting efficiency. Due to the coupling effect and high performance of the hybrid energy-harvesting systems, they have drawn considerable attention as a potential candidate for sustainable/renewable energy harvesting. Examples of hybrid energy harvesters (HEHs) include a combination of mechanical and photovoltaic energy harvesters [42–47], mechanical and thermal energy harvesters [48–52], thermal and photovoltaic energy harvesters [53–55], and combinations of other energy harvesters [56–60].

### 2.1. Piezoelectric–Electromagnetic Hybrid Energy Harvesters

Piezoelectric and electromagnetic processes are frequently utilized to produce electricity from kinetic energy. These processes are integrated in the HEHs to increase the system power density and the potential to generate more energy [61–69]. One of the goals of these solutions is to improve the electrical dampening and match it with the mechanical one to increase the efficiency of energy conversion in the HESs. Xia et al. created a novel approach for the piezoelectric–electromagnetic (PE-EM) harvesters by altering the axial magnetic force, as illustrated in Figure 2a [70]. The results showed a broad operating frequency range

of about 25.5–62 Hz. In response to this, in the cantilever harvester, Xu et al. [71] added another magnetic oscillator in between the coils and tip magnet, as shown in Figure 2b. Power control circuits are a challenge for hybrid energy harvesting. Output in the form of alternating currents is common for the harvesters utilizing piezoelectric and electromagnetic technologies. It is necessary to convert this alternating current into a more stable form through rectification, storing the energy and stabilizing the voltage to accumulate the charges collected in a single storage unit. Piezoelectric energy harvesters (PEHs) often have high output impedance due to their low capacitance and operating frequency. Conversely, the electromagnetic harvester yields high output current and low voltage due to its lower impedance in the coil. The distinct variations in output characteristics between PEHs and electromagnetic energy harvesters pose considerable challenges when designing an effective interface for hybrid harvesters. A practical approach involves designing separate rectification and storage components for piezoelectric and electromagnetic harvesters and operating them concurrently, as illustrated in Figure 2c [72]. The structural design of PE-EM HEHs has been well studied. Still, one of the problems is that there is not enough synergy to maximize the benefits of both the electromagnetic and piezoelectric conversion in a single hybrid design. Most recent works either focus on exploring additional non-linear dynamics that might be applicable to non-hybrid systems or separately adding the two conversions. So, to make the PE-EM HEHs more advantageous, finding more efficient ways to control two power sources is essential.

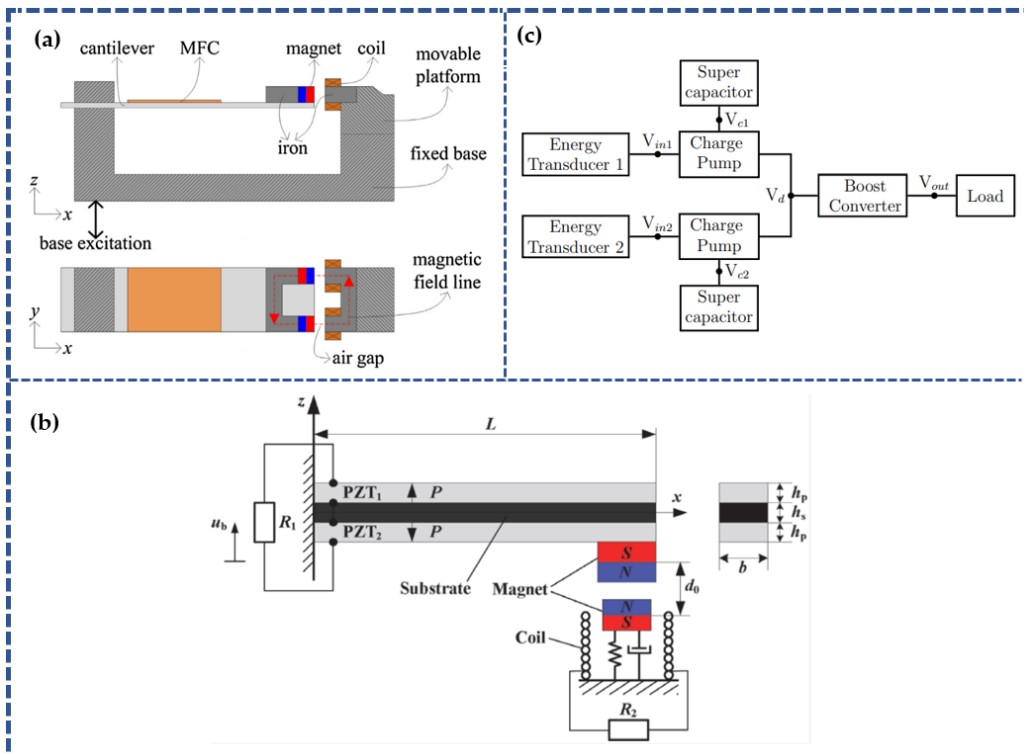

**Figure 2.** Schematic diagram of PE-EM hybrid energy harvester proposed by (**a**) Xia et al. [70]; (**b**) Xu et al. [71]; (**c**) proposed power management circuit for controlling power in hybrid PE-EM systems [72].

## 2.2. Piezoelectric–Triboelectric Hybrid Energy Harvesters

The piezoelectric effect and the triboelectric effect are mechanisms for converting mechanical energy into electrical energy through the fundamental concept of displacement current. When two triboelectric materials come into contact or separate from one another under the influence of an external force, electrons are transferred, and the triboelectric nanogenerator (TENG) creates a potential difference over the surfaces of the materials due

to the flow of current. In contrast, the piezoelectric energy harvester creates an internal electrical potential across the piezoelectric material. When a stress is applied on the piezoelectric material, this stress causes a displacement of electric charges within a material, resulting in an electrical potential and subsequent electrical current. Hence, these two effects exhibit certain similar operational properties in responding to mechanical vibration, compression, and deflection, which may be further incorporated as the hybrid energy-harvesting system for increasing the energy output. Recent revolutionary studies have demonstrated that piezoelectric and triboelectric phenomena can coexist in a particular function material and interact, opening a new path to improve the performance of hybrid piezoelectric–triboelectric (PE-TE) devices. Nanoparticles of ferroelectric barium titanate (BTO) were combined with a polydimethylsiloxane (PDMS) by Suo et al. [73] to create a composite film of BTO/PDMS that outperformed the pure film of PDMS. It has been found that adding BTO nanoparticles with positive polarization will increase the piezoelectricity's contribution. Han et al. [74] demonstrated a wide band and low-frequency hybrid harvester that used a variety of piezoelectric PVDF cantilevers to periodically impact the bottom triboelectric PDMS layer to generate both triboelectric and piezoelectric outputs. The working principle of this PE-TE hybrid energy harvester is shown in Figure 3A.

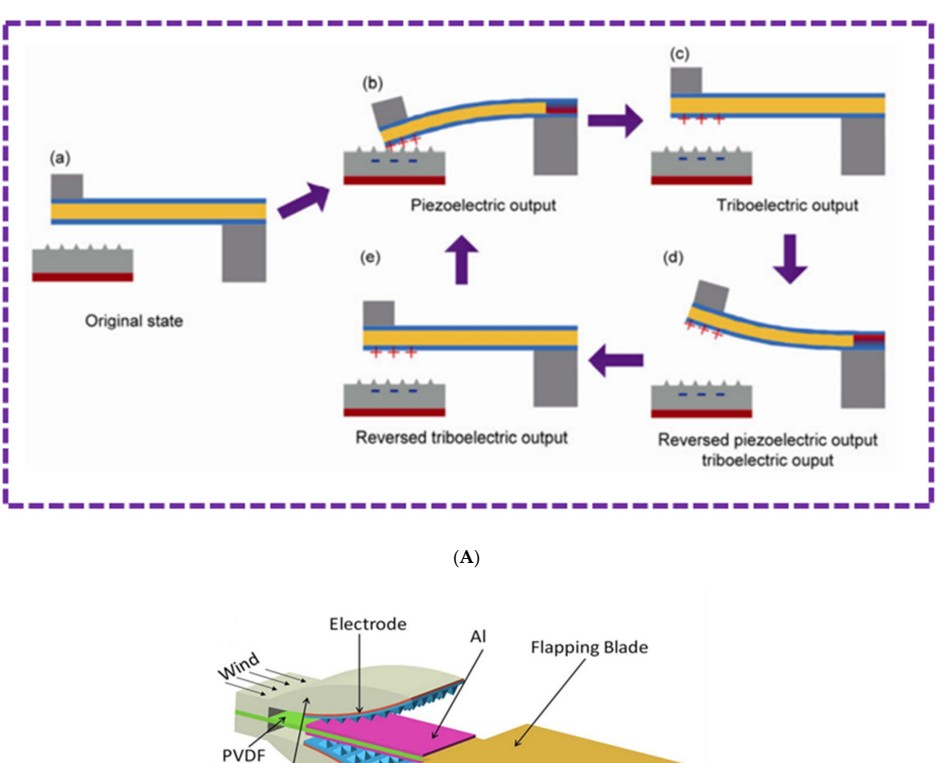

**Figure 3.** (**A**). Working mechanism of the PE-TE hybrid energy harvester: (**a**) initial condition; (**b**) the cantilever bends and contact the MNDS PDMS surface, generating a piezoelectric effect; (**c**) the cantilever recovers, enlarging the gap, resulting in a triboelectric effect; (**d**) the cantilever bends in an alternative direction, widening the gap, leading to a triboelectric and reversed piezoelectric effect; (**e**) the cantilever recovers, reducing the gap, resulting in a reversed triboelectric effect [74]. (**B**). Schematic diagram of PE-TE hybrid wind energy harvester proposed by Chen et al. [75].

Moreover, some wind harvesters combine the phenomena of piezoelectricity and triboelectricity as well simultaneously. Hybrid wind energy harvesting that is based on the vortex shedding phenomenon was proposed by Chen et al. [75] and is shown in Figure 3B. A PVDF cantilever's output and triboelectric effect can be maximized by the appropriate design of the flapping blades and the spindle-shaped frames.

### 2.3. Electromagnetic–Triboelectric Hybrid Energy Harvesters

To benefit from electromagnetic and triboelectric energy harvesters, several researchers tried to integrate TENG and electromagnetic energy harvesters (EMEHs) into one hybrid device. The EMEH and TENG, however, are unable to effectively exchange power processing circuits. To harvest wind energy, Wang et al. [76] created a hybrid wind energy harvester that combines both EMEH and TENG. When the central oscillating FEP film is oscillating vertically by the wind flow, it contacts both the upper and lower electrodes, causing an electron flow in the TENG component. Additionally, the varying distance between the oscillating magnets on the central film and coils on the upper and lower bases enables the EMEH component to produce output voltage or current simultaneously. A water wave energy-harvesting system comprising the EMEH and TENG components was proposed by Wang et al. [77], as shown in Figure 4a. A series of aluminum (Al) rolling rods and polytetrafluoroethylene (PTFE) film covered with the copper inter-digital electrodes make up the TENG component. Four steel rods were placed between the top and bottom magnet arrays in the EMEH component to direct the copper coil motion. The hybrid generator simultaneously enables a simultaneous increase in the operating frequency range and maximizes the energy conversion efficiency at a low frequency below 1.8 Hz. A wearable hybrid electromagnetic–triboelectric harvesting wristband utilizing relatively low wrist motion was presented by Maharjan et al. [78,79], as depicted in Figure 4b. A magnetic ball that could move freely inside of a hollow tube was used in the device. The production of nanorods on the inner side of the tube, microstructures over the magnetic ball, and flux-concentrated material combined with the coil significantly increased the overall output performance. To capture the rotational energy, (Chen et al. [80] and Zhang et al. [81]) have created a rotating disc that is based on EM-TE hybrid generators. The rotating mechanism comprises a rotor and stator, facilitating the creation of relative motion between the positively and negatively charged triboelectric materials, along with the magnets and coils.

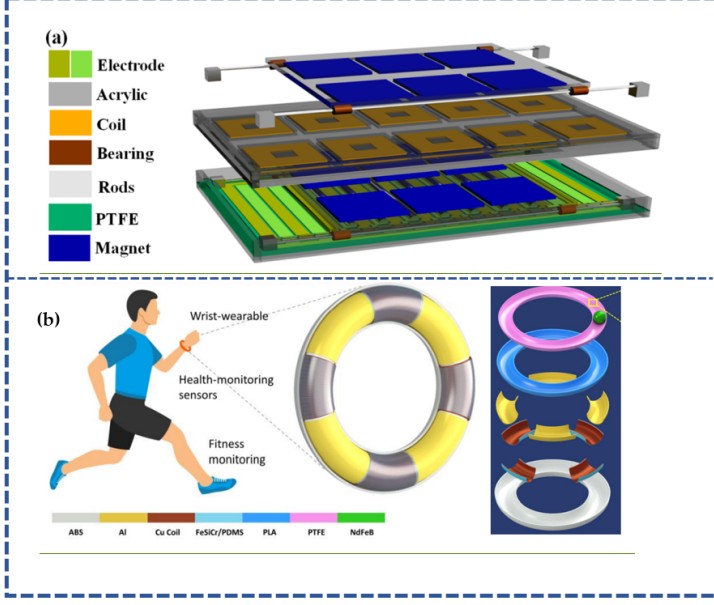

**Figure 4.** (**a**) Schematic diagram of water wave hybrid energy harvester proposed by Wang et al. [77]; (**b**) wearable hybrid electromagnetic–triboelectric harvesting wristband by Maharjan et al. [79].

### 2.4. Piezoelectric–Electromagnetic–Triboelectric Hybrid Energy Harvesters

Researchers are looking at the possibilities of triple hybrid energy-harvesting technologies based on dual hybrid systems combining PE-TE, EM-TE, and PE-EM mechanisms. Combining the piezoelectric–electromagnetic–triboelectric (PE-EM-TE) harvesting systems into one device may be a potential way to further enhance the output performance. A hybridized PE-EM-TE generator using a central magnet floating structure with increased vibrational sensitivity was described by et al. [82]. The peak power (below 20 Hz), produced by the bottom EMEH was 38 mW, and that from the top EMEH was 36 mW. The peak power produced by the bottom PEH was 105 mW and that from the top PEH was 122 mW. Compared with these components, TENG produced a negligible peak power of about 78 μW. Koh et al. [83] exhibited a self-powered inertial sensor containing non-resonant magnetic balls that move within a hollow shell, as depicted in Figure 5. The interior surface of the shell was layered with PTEF, PVDF, and Al films, while wire coils were wound around the exterior. This setup allowed the utilization of PE-EM-TE hybrid effects to harness energy from three-dimensional vibrations, rotation, and unpredictable human movements. The device was proven to detect acceleration in the *x*, *y* and *z* directions, as well as angular velocity in roll, pitch, and yaw axes, displaying potential application in healthcare monitoring for recognizing human motion.

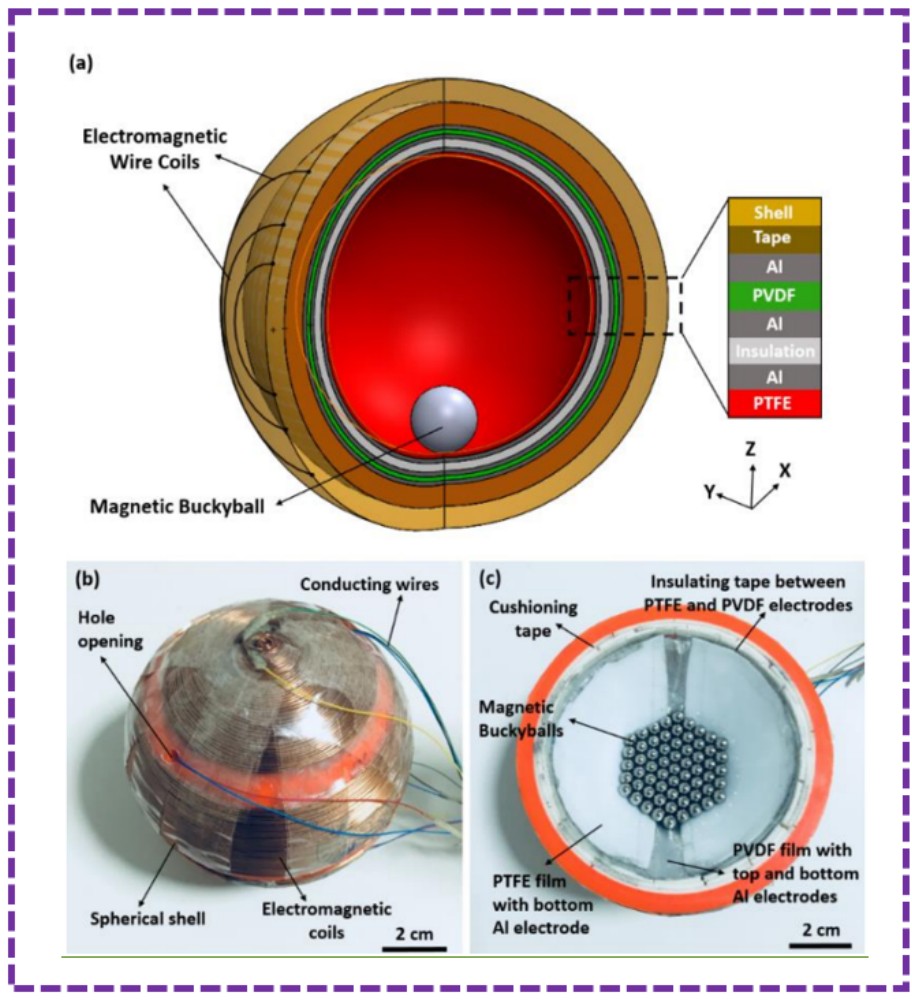

**Figure 5.** (**a**) Schematic diagram of the hybrid PE-EM-TE system proposed by Koh et al. [83]; (**b**) outside view; (**c**) inside view [83].

### 2.5. Various Hybrid Energy-Harvesting Systems

In most situations, energy sources like heat, vibrations, and light coexist; however, some of these may only be partially accessible or available. Consider humans as an illustrative example. People will move a lot when they are traveling or exercising but not much when they are at rest. An alternative would be to use other sources of energy, such as light or thermal energy. Hence, researchers are looking into hybrid energy-harvesting methods that combine different sources of energy into one device to provide a resilient and sustainable power supply [84–86]. To harvest different sources of energy, Gambier et al. created an HEH comprising layers of flexible solar panels, thin film batteries, a thermoelectric generator and piezoceramic [87]. A combined power management circuit-equipped hybrid harvester from thermal and indoor light energy was presented by Tan and Panda [84]. A total of 620 μW of output power was collected. By utilizing both kinetic energy and light energy, Chen et al. developed a foldable, flexible, and lightweight hybrid energy-harvesting technology for wearable applications [88]. To develop the smart fabric having a thickness of about 320 μm, photovoltaic textile and fiber-based triboelectric generators were integrated. The output power of this (hybrid energy harvesting) textile, when exposed to sunlight, human motion, and wind, was around 67 μW [89].

These various transduction mechanisms as described in this section have various advantages and disadvantages. A PE-EM hybrid energy harvester combines high energy conversion efficiency by capturing both mechanical vibrations and electromagnetic induction, but there are design complexities and challenges in power management. PE-TE offers versatility by harvesting energy from two different sources. However, it faces challenges related to material compatibility and potential losses due to friction. EM-TE is prone to mechanical wear and tear due to its operating mechanism. PE-EM-TE has superior energy-harvesting capabilities through multiple mechanisms, yet the design complexity and system optimization present significant drawbacks. Similarly, thermoelectric, and piezoelectric optimization for varying temperature gradients and material properties are significant challenges.

Table 1 below shows the performance comparison of various hybrid energy harvesters (HEHs) reported in the literature based on their configuration, energy sources and output performance.

**Table 1.** Performance comparison of various hybrid energy harvesters.

| Hybrid Energy-Harvesting System | Configuration | Energy Source/Device Size | Output Performance | References |
|---|---|---|---|---|
| **Piezoelectric–Electromagnetic Energy Harvesters** | | | | |
| PE-EM; PZT disc, coil/magnet in approaching separation mode | Helmholtz resonator | 20 × 24 mm | PEH: 49 μW EMEH: 3.2 μW | Khan et al. [90] |
| PE-EM; PZT bimorph, coil/magnet in horizontal sliding mode | Airfoil and cantilever Flow induced vibration | - | No prototype developed | Dias et al. [91] |
| PE-EM; Fixed PZT beam, coil/magnet in horizontal sliding mode | Oscillating magnet Force amplification | 70 × 45 × 20 mm | 0.33 W peak power | Li et al. [92] |
| PE-EM; PZT disc, coil/magnet in approaching separation mode | Free sliding magnet | 68 × 39 × 37 mm | 50–130 μW | Hamid et al. [93] |
| PE-EM; Bimorph PZT, coil/magnet in approaching separation mode | Multi-modal oscillations | 50 × 20 × 80 mm | 1.4 mW at 22.6 Hz | Xu et al. [71] |
| PE-EM; PZT-coated d31, coil/magnet in forthcoming horizontal sliding and separation mode | Cantilever resonance | 22 × 10 × 10 mm | PEH: 176 μW EMEH: 0.19 μW | Yang et al. [94] |
| PE-EM; PZT patch, coil/magnet in approaching separation mode | Airfoil and cantilever Dual beam structure | 85 × 80 × 40 mm | PEH: 156 μW EMEH: 1.57 mW | Iqbal et al. [95] |
| PE-EM; PZT stack, coil/magnet in approaching separation mode | Oscillating magnet Tri-stable | - | No prototype developed | Yang et al. [96] |

**Table 1.** *Cont.*

| Hybrid Energy-Harvesting System | Configuration | Energy Source/Device Size | Output Performance | References |
|---|---|---|---|---|
| PE-EM;<br>PZT cantilever, coil/magnet in approaching separation mode, non-linear levitation | Free sliding<br>Multi-directional<br>Dual-stable | 14 × 55 mm | EMEH: 1.23 mW<br>PEH: 0.18 mW | Fan et al. [97] |
| PE-EM;<br>PZT coating applied to Al, coil/magnet in approaching separation mode | Multi-mode vibration | 93 × 30 × 15 mm | PEH: 250 μW<br>EMEH: 244 μW | Toyabur et al. [98] |
| PE-EM;<br>PZT bimorph, coil/magnet in approaching separation mode | Fixed-fixed beam<br>Mono-stable | 50 × 10 × 15 mm | No result provided | Mahmoudi et al. [99] |
| PE-EM;<br>PZT-coated d33 cantilever, coil/magnet in approaching separation mode | Cantilever resonance | 44 × 24 × 30 mm | 332 μW at 21.6 Hz | Challa et al. [100] |
| **Piezoelectric–Triboelectric Energy Harvesters** | | | | |
| PE-TE;<br>Al/PVDF/Al, Al/PDMS/Al | Flapping blade | Wind flow | PEH: 112 μW<br>TENG: 76 μW | Chen et al. [75] |
| PE-TE;<br>Ag/PZT-5J/Ag, Al/PTFE/Nylon/Al | Truss stopper | Vibration | PEH: 14 mW<br>TENG: 5.7 mW | Li et al. [101] |
| PE-TE;<br>Conductive fabrics, fibroin/PVDF nanofiber | Laminate | Pressing force | 0.31 mW/cm$^2$ | Guo et al. [102] |
| PE-TE;<br>Au/ZnO NFs + PDMS/Ni + 3D Gr | Laminate | Pressing force | 6.22 mW/cm$^2$ | Qian et al. [103] |
| PE-TE;<br>PET/ITO/BTO + PDMS/Cu | Laminate | Pressing force | No result provided | Suo et al. [73] |
| PE-TE;<br>Al/PVDF/Al,<br>Al/PDMS/MWCNT-PDMS/Au | Parallel plate | Pressing force | PEH: 2.27 × 10$^{-3}$ mW/cm2<br>TENG: 2.04 × 10$^{-3}$ mW/cm$^2$ | Zhu et al. [104] |
| PE-TE;<br>AZO/P(VDF-TrFE)/AZO,<br>AZO/PDMS/Skin | Laminate | Pressing force | 0.075 mW/cm$^2$ | Wang et al. [105] |
| PE-TE;<br>Au/P(VDF-TrFE)/Au/P(VDF-TrFE)/Au,<br>Al/PTFE/Au | Rotational blade | Rotation | 10.88 mW | Zhao et al. [106] |
| PE-TE;<br>Al/PVDF/Al, Al/PDMS/ITO | Cantilever stopper | Vibration | No result provided | Han et al. [74] |
| PE-TE;<br>Al/PTFE + PVDF + PDMS/Li-ZnO + MWCNT/Ag | Laminate | Pressing force | No result provided | Chowdhury et al. [107] |
| PE-TE;<br>Cu/ZnO + MWCNT + EGO + PDMS/Cu | Laminate | Pressing force | No result provided | Karumuthil et al. [108] |
| PE-TE;<br>Al/PVDF/Al, Al/PDMS/ITO | r-shape | Pressing force | PEH: 10.95 mW/cm$^3$<br>TENG: 2.04 mW/cm$^3$ | Han et al. [109] |
| PE-TE;<br>Cu/PTFE/PVDF/Cu, Cu/PTFE/Cu | Parallel plate | Pressing force | PEH: 0.15 mW/cm$^2$<br>TENG: 2.75 mW/cm$^2$ | Zhu et al. [110] |
| PE-TE;<br>Au/PVDF/Au, Au/PTFE/Al | Arc shape | Pressing force | 4.44 mW/cm$^2$ | Jung et al. [111] |
| **Electromagnetic–Triboelectric Energy Harvesters** | | | | |
| EM-TE;<br>Coil/magnet in horizontal sliding mode, Cu/FEP in lateral sliding mode | Rotating sleeve | Rotation, wind flow | 13.8 μW/cm$^3$ | Cao et al. [68] |
| EM-TE;<br>Coil/magnet in horizontal sliding mode, Au/PTFE in freestanding triboelectric layer mode | Rotating disk | Rotation | EMEH: 176.9 μW/cm$^3$<br>TENG: 111.6 μW/cm$^3$ | Chen et al. [80] |
| EM-TE;<br>Coil/magnet in horizontal sliding mode, Cu/Silicone in freestanding triboelectric layer | Magnetic roller laterally | Water wave | EMEH: 39.4 μW/cm$^3$<br>TENG: 0.21 μW/cm$^3$ | Hao et al. [112] |
| EM-TE;<br>Coil/magnet in horizontal sliding mode, Cu/PTFE in freestanding triboelectric layer | Magnetic slider laterally | Water wave | EMEH: 1.32 μW/cm$^3$<br>TENG: 1.05 μW/cm$^3$ | Wang et al. [77] |
| EM-TE;<br>Coil/magnet in approaching separation mode, Al/PDMS in lateral sliding mode | Magnetic slider laterally | Vibration, human motion | 381 μW/cm$^3$ | Salauddin et al. [113] |
| EM-TE;<br>Coil/magnet in approaching separation mode, Al/FEP in contact separation mode | Spring-mass vertically | Vibration | EMEH: 9.7 μW/cm$^3$<br>TENG: 37.6 μW/cm$^3$ | Liu et al. [114] |
| EM-TE;<br>Coil/magnet in horizontal sliding mode, Cu/PTFE in freestanding triboelectric layer | Rotating sleeve | Rotation, wind flow | EMEH: 0.34 mW<br>TENG: 2.13 mW | Qian et al. [115] |
| EM-TE;<br>Coil/magnet in approaching separation mode, Cu/FEP in contact separation mode | Vibration film vertically | Wind flow | EMEH: 58.3 μW/cm$^3$<br>TENG: 39.6 μW/cm$^3$ | Wang et al. [76] |

**Table 1.** *Cont.*

| Hybrid Energy-Harvesting System | Configuration | Energy Source/Device Size | Output Performance | References |
|---|---|---|---|---|
| EM-TE; Coil/magnet in horizontal sliding mode, Cu/Silicone in freestanding triboelectric layer | Magnetic roller laterally | Water wave | EMEH: 9 mW EMEH: 0.8 mW | Wang et al. [116] |
| EM-TE; Coil/magnet in approaching separation mode, Al/PTFE in freestanding triboelectric layer | Magnetic ball laterally | Human motion | EMEH: 5.14 mW/cm$^3$ TENG: 0.22 μW/cm$^3$ | Maharjan et al. [78] |
| EM-TE; Coil/magnet in lateral sliding mode, Cu/Kapton in contact separation mode | Spring-mass laterally | Vibration, human motion | EMEH: 29.9 μW/cm$^3$ TENG: 0.78 μW/cm$^3$ | Chen et al. [117] |
| EM-TE; Coil/magnet in approaching separation mode, ITO/PTFE in contact separation mode | Spring-mass vertically | Vibration | 1.3 μW/cm$^3$ | Gupta et al. [118] |
| **Piezoelectric–Electromagnetic–Triboelectric Energy Harvesters** | | | | |
| PE-EM-TE; PVDF in bending, coil/magnet in horizontal sliding mode, Al/PTFE in lateral sliding and contact separation mode | Rotating sleeve and disk | Wind flow | PEH: 1.38 mW EMEH: 268.6 mW TENG: 1.67 mW | Toyabur Rahman et al. [119] |
| PE-EM-TE; Compressed state PZT sheet, coil/magnet in approaching separation mode, Ni/silicone in single electrode mode | Magnetic mass stopper | Vibration | PEH: 122 mW, EMEH: 38.4 mW TENG: 78.4 μW | He et al. [82] |
| PE-EM-TE; Compressed PVDF Sheet, coil/magnet in approaching separation mode, Al/PTFE in freestanding triboelectric layer mode | Magnetic rolling ball | Human motion | PEH: 0.19 μW, EMEH: 22.4 nW TENG: 0.72 μW | Koh et al. [120] |
| PE-EM-TE; Bending state PVDF sheet, coil/magnet in approaching separation mode, Cu/PVDF in contact separation mode | Spring mass stopper | Vibration | PEH: 41 μW EMEH: 66.5 μW TENG: 4.6 μW | He et al. [121] |
| **Various Other Hybrid Energy Harvesters** | | | | |
| Piezoelectric with magnet | - | Vibration and magnetic; 150 × 30 × 1 mm | 50 μW | Xu et al. [122] |
| Piezoelectric and pyroelectric | - | Thermal and vibration; 70 × 10 × 0.7 mm | 0.4 μW | Kang et al. [123] |
| Photovoltaic (PV) and radio-frequency (RF) | - | Solar and electromagnetic (EM); EM: 47 × 47 × 20 mm PV: 114 × 24 mm | PV: 93 mW RF: 28 μW | Bito et al. [124] |
| Piezoelectric (PE), photovoltaic and thermoelectric generation (TEG) | - | Vibration, light and thermal; 93 × 25 × 1.5 mm | TEG: 6.6 mW PV: 12.5 mW PE: 0.49 mW | Gambier et al. [87] |
| Piezoelectric and pyroelectric | - | Thermal and vibration; layer thickness 0.7 μm | 400 mV | Lee et al. [125] |
| Electromagnetic (EM), thermoelectric generation (TEG) and piezoelectric (PE) | - | Electromagnetic, thermal and Vibration; EM: 140 × 20 × 50 mm TEG: 500 × 82 × 10 cm PE: 90 × 17 × 0.8 mm | EM: 0.7–366 mW TEG: 12.9 mW to 1.98 W PE: 0.63 mW | Yang et al. [126] |
| Triboelectric and photovoltaic | - | Solar and mechanical; 50 × 40 × 0.32 mm | 0.5 mW | Chen et al. [88] |
| Photovoltaic and thermoelectric generation | - | Light and thermal; PV: 55 × 30 × 1 mm TEG: 20 × 20 × 20 mm | 621 μW | Tan et al. [84] |

## 3. Applications of Sustainable Hybrid Energy Harvesting

To attain self-sustaining electronics, Internet of Things (IoT) devices, and self-powered smart wireless sensor nodes systems in a variety of applications, it is feasible to use renewable/sustainable hybrid energy sources, such as heat, light, wave, wind, human motion, vibration, radio frequency, radiations, and bioenergy. These sources offer potential and long-term solutions. This section explores a range of applications for HEHs and delves into the characteristics of different energy sources, the technologies that are used for energy collecting, and the effect they have. The characteristics of energy sources vary widely. Various technologies are employed to capture energy effectively. These include photovoltaic cells for harnessing solar energy and piezoelectric and electromagnetic converters for converting mechanical vibrations into electricity. Each technology is suited to specific energy characteristics. Piezoelectric and electromagnetic technologies are particularly effective in harvesting energy from vibrations with applications in industry condition

monitoring and human healthcare. Some of the applications of HEHs are discussed in this section below, and a flowsheet diagram of applications is shown in Figure 6.

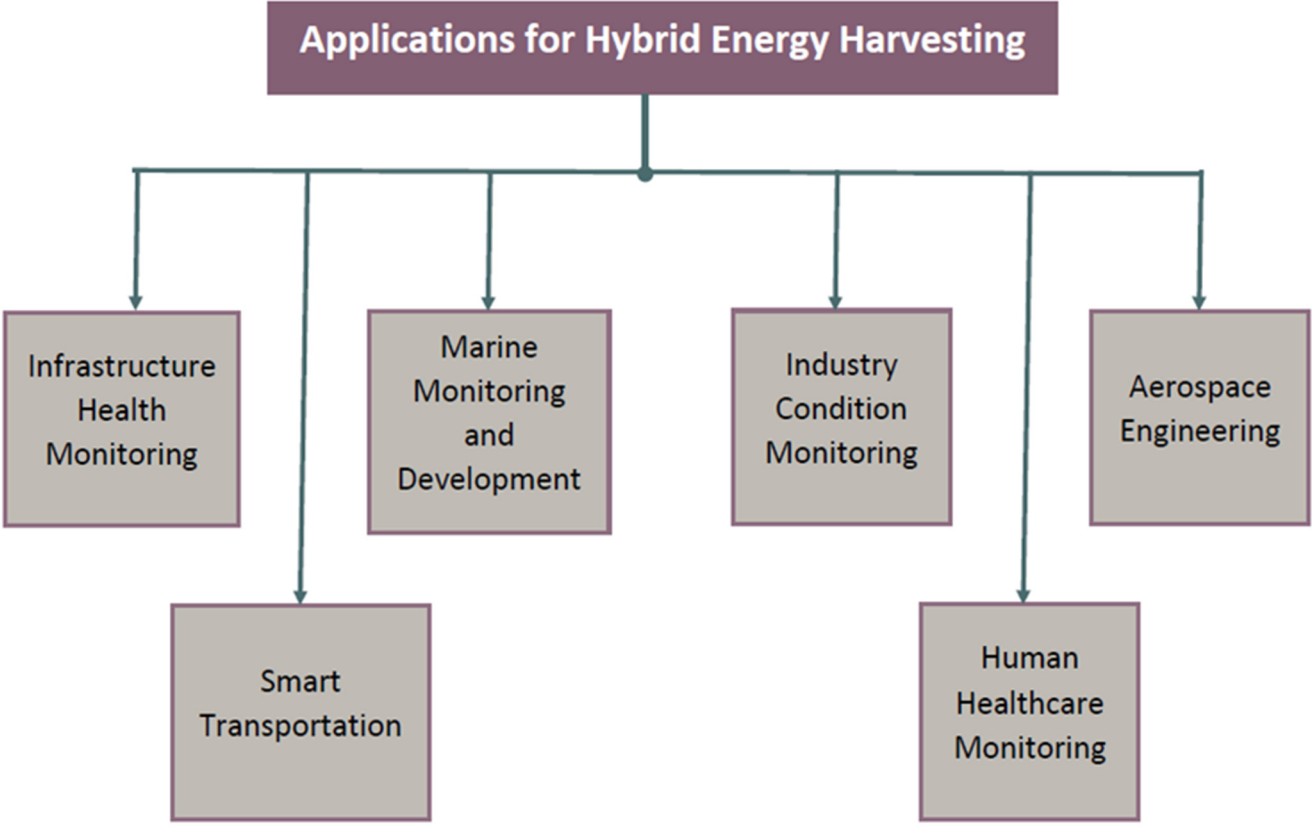

**Figure 6.** Flowchart of applications of hybrid energy harvesting.

*3.1. Smart Transportation*

The latest trends in transportation, particularly in the automobile sector, are automated vehicles and electrification. To meet the demands of digitalization and automation, electronics and sensors will be incorporated into the structures on a significantly bigger scale. A more reliable and effective power supply can be achieved by utilizing various forms of energy sources. Many different energy sources can be used to power vehicles, including trains, cars, ferries, airplanes, and buses. Utilizing integrated multi-mode vibrations and mechanical non-linearity, a broadband vibration-based energy harvester was designed for the self-powered monitoring systems of the underground trains by Fu et al. [127]. The idea of harnessing wind energy through the aerodynamic losses on the highways has been covered by numerous recent patents globally [128]. The GPS and accelerometer are the main signal sources in the current systems, which quickly consume the device's battery [129]. Hybrid energy-harvesting technologies may be capable of providing transportation vehicles with self-powered sensing capabilities for event detection and condition monitoring. In general, it is well recognized that energy can be transferred from one form to another, as also seen in Figure 7 [130]. In this way, mechanical energy is converted to electrical energy using piezoelectric and electromagnetic technology (Figure 7). Many studies have been conducted in this field on energy harvesters utilizing single sources of energy or single conversion processes [116]. Yet, there is limited information and few examples of hybrid systems. In contrast to certain applications, like biomedical devices, airflow and vibration are abundant in the transportation systems at comparatively high-frequency ranges and energy levels, and the size limitation is frequently less demanding. Piezoelectric and electromagnetic-based conversion methods are suitable. Triboelectric methods can also be utilized in situations like tire pressure monitoring; however, one of the challenges is

material reliability. The utilization of other energy sources like solar and thermal is also possible, but their mounting restrictions are more severe.

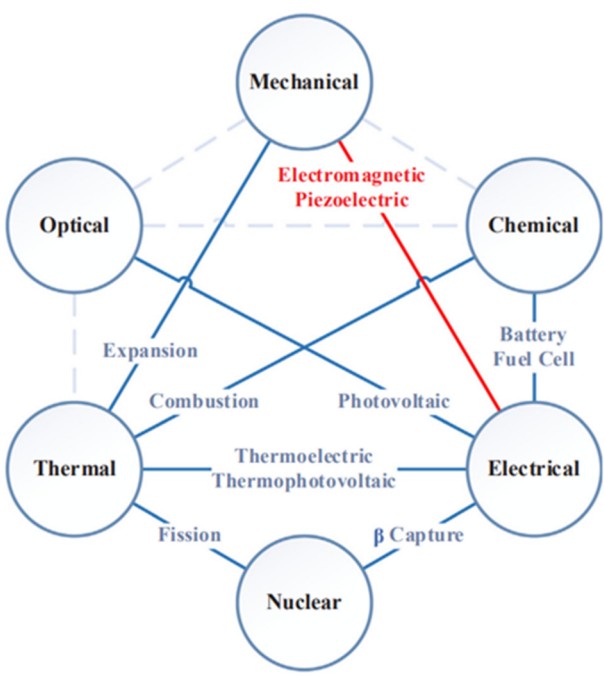

**Figure 7.** Flowchart of transformational processes among various energy types [130].

### 3.2. Infrastructure Health Monitoring

There is widespread agreement regarding the significance of managing and monitoring the condition of civil infrastructures, such as water management, power and communication infrastructure, roads, buildings, bridges, railways, tunnels, environmental monitoring, and agricultural facilities. Energy harvesting is the potential method that helps to generate clean, renewable energy and increase the sustainability of infrastructures. Figure 8a shows each energy-harvesting technology that is available and can be used on roadways. An energy generator, storage device, and electrical circuit are the three main parts of energy-harvesting systems. The energy generator transforms thermal, mechanical, and solar energy from the environment into electrical energy. The resulting voltage is then increased and regulated by the electrical circuit to assist in making it suitable for a variety of applications. The energy that has been captured is then saved for subsequent use in supercapacitors or rechargeable batteries. The amount of energy produced by an energy-harvesting system might vary greatly depending upon the principle underlying the harvesting technology used. The amount of energy produced overall depends on various factors, including the availability and intensity of the environmental energy source and the effectiveness of the conversion process. By using HEHs, it significantly increases the output performance by converting the mechanical energy into electrical using multiple transduction mechanisms [131]. To monitor the health of an infrastructure, a high proportion of WSNs are arranged, which enables continuous detection that may eventually save lives as well as minimize downtime and economic losses. As the infrastructural systems are generally located outdoors, a variety of renewable energy sources, including solar, wind, rain, and radio-frequency energy, are readily available instead of traditional batteries and wire power supplies. In recent decades, the regular frequency of natural disasters has posed a serious threat to human lives and property. A self-powered hybridized electromagnetic–triboelectric (EM-TE) harvester and a solar cell for monitoring the state of natural disasters were reported by Qian and Jing [115], as shown in Figure 8b. To monitor earthquakes and detect fires, temperature and vibration sensors were used, respectively. A revolving, wind-driven hybridized energy harvester (WH-EH) can be incorporated using

the WSN technology to create a self-sustaining global disaster-monitoring device. The rotator in this harvester is directly powered by the external rotational motion, making it simple to combine the TENG with 18 electromagnetic generators. A thermoelectric and electromagnetic HEH was created by Liu et al. [132] using the fluid velocity and temperature change in an irrigation system, as shown in Figure 8c. Different monitors or sensor nodes, such as flow meters and temperature sensors, can be combined with power source sources from renewable energy harvesters in a smart agriculture irrigation system. The turbine-fan with magnets attached to the blades is positioned within the water pipe to harness the motion of water flow. The energy due to water flow is converted into electricity by multiple coils as the turbine rotates. To capture energy from the temperature differences, thermoelectric generators are installed around the pipe. To enable long-term self-powered wireless sensor applications for intelligent agriculture, smart buildings, structural health monitoring, environmental monitoring, security, and facility monitoring, and so on, further research and development is still required.

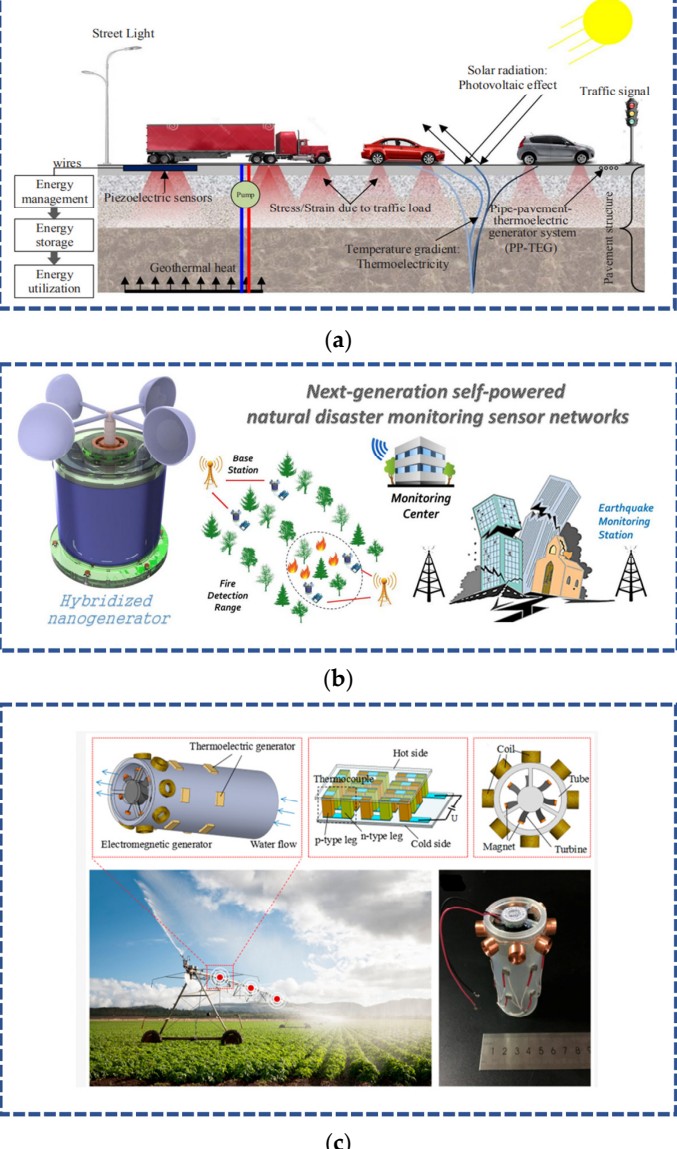

**Figure 8.** (**a**) Sources for energy harvesting that are accessible in roadways [131]; (**b**) a revolving WH-EH incorporating using WSN technology for natural disaster monitoring developed by Qian and Jing [115]; (**c**) hybrid energy harvester in irrigation system developed by Liu et al. with thermoelectric generators installed around the pipe [132].

### 3.3. Marine Monitoring and Development

Different countries and research groups are paying more attention to marine monitoring and development. The power supply, and more particularly the carrying battery's capacity, has a significant role in limiting the functional life of the marine equipment. It has been recognized that the ocean contains a significant quantity of renewable energy, including wind energy, solar energy, water wave energy, tidal energy, temperature gradient energy, salinity gradient energy, and water flow energy [133,134]. The blue-energy-harvesting technique that is based on the different transduction mechanisms has drawn the attention of many researchers in recent years. These methods were discovered to be a potential source for self-charging batteries or self-powered sensors in marine monitoring and development systems [135,136]. For the sustainable growth of society, blue energy, which is obtained from the ocean waves, is a significant and promising renewable energy source. Both TENGs and EMGs are recognized as potential methods for harnessing blue energy. Wu et al. [137] present a hybridized TE-EM water-wave energy harvester (WWEH) that is based on the magnetic sphere, as shown in Figure 9a. A freely rotating magnetic sphere detects the water's motion to move the friction element for the TENG backward and forth on the solid surface. By the electromagnetic induction phenomenon, two coils simultaneously convert the movement of a magnetic ball/sphere into electricity. This work illustrates that distributed self-powered environmental-monitoring sensors can be driven successfully by the WWEH. Similarly, a hybridized TE-EM WWEH, based on a chaotic pendulum, is presented by Chen et al. [138], as illustrated in Figure 9b. The major pendulum and the inner pendulum are the two parts that make up this chaotic pendulum. The main pendulum simply swings back and forth in time with the oscillations of the water waves. The inner pendulum, which has three magnetic balls that are evenly spaced out on a revolving shaft, however, moves chaotically and unexpectedly. An electromagnetic nanogenerator (EMG) and triboelectric nanogenerator (TENG) are the two components of the hybridized nanogenerator. The central pendulum is connected to the TENG, which is made of gold electrodes and polytetrafluoroethylene (PTFE) films. Through the friction that occurs between the PTFE layer and the interdigitated electrodes, oscillating mechanical energy is transformed into electrical energy. Inside the chaotic pendulum, the EMG comprises three magnetic balls and three coils. The magnetic flux among the copper coils changes as the magnetic balls move under the influence of external magnetic forces and gravity, producing electrical energy. The harvester's physical design made use of the chaotic pendulum's high electromechanical conversion ratio and low working frequency. Wang et al. [116] presented a ship-shaped hybridized nanogenerator (SHNG) made up of three triboelectric nanogenerators and an electromagnetic generator, as shown in Figure 9c. Because of the less frictional resistance created by rolling the magnetic cylinder in this design, the TENG can easily be powered by a water wave.

This study not only offers a novel approach for efficiently harvesting blue energy but also offers a significant opportunity for enabling self-powered marine rescue devices and self-desalination. The combination of water-wave power generators, solar panels, and wind turbines can create a generation power network over the ocean's surface, according to the blue energy dream outlined by Wang et al. [139], as shown in Figure 9d. On a floating platform, the electricity generated by wind-powered turbines, solar cells, and TENG networks could be utilized locally or sent to onshore electric grids or power plants. For marine monitoring and development, hybridized blue energy-harvesting technology can hold significant value in the self-powered Internet of Things.

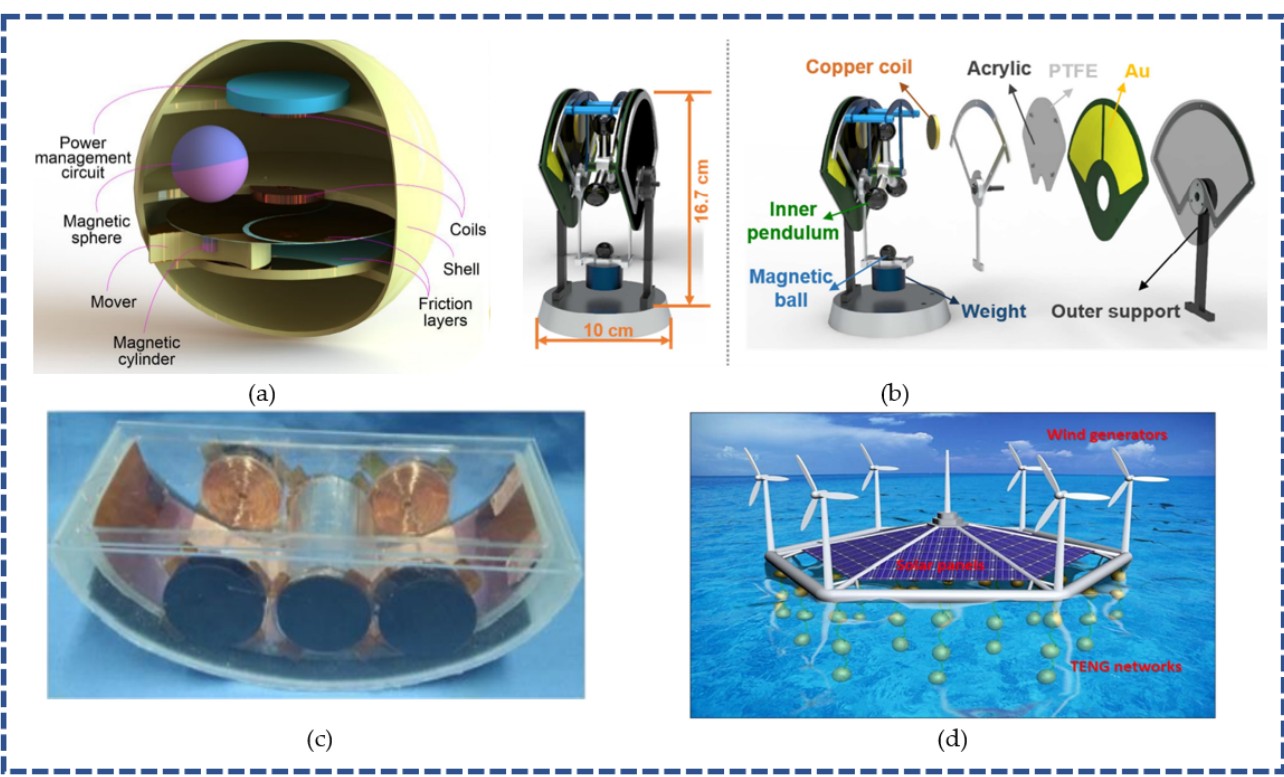

**Figure 9.** (**a**) TE-EM WWEH schematic diagram containing a freely rotating magnetic sphere that detect the water motion, as presented by Wu et al. [137]; (**b**) TE-EM WWEH based on chaotic pendulum schematic diagram presented by Chen et al. [138]; (**c**) SHNG having a rolling cylinder developed by Wang et al. [104]; (**d**) blue energy dream outlined by Wang et al. [139].

### 3.4. Human Healthcare Monitoring

The advancement of implantable and wearable electronic devices can help professionals intervene in chronic illnesses as soon as possible. The sustainable supply of power is one of the major constraints. A human body may have a variety of energy sources available, such as muscle contraction, human motion, body heat, cardiac and lung motions, blood pulsation, etc. Several energy-harvesting designs have been proposed for replenishing batteries and ultimately developing self-powered implantable or wearable electronic devices to harvest the human body energies [140–143]. Zhu et al. [144] developed self-functional socks that provide self-powered monitoring and the sensing of different physiological signals, like contact force, sweat level gait, etc. They did this by utilizing the hybrid PE-TE energy harvester phenomena from human walking, as shown in Figure 10a. With a frequency of 2 Hz and a load resistance of 59.7 MΩ, an output power of 1.71 mW is obtained. All the described conversion methods have advantages at various sources or areas of the human body. For instance, the mechanical energy that is provided by the footsteps is considerable, making this suitable for TE and EM [145–147]. For applications involving the upper body and the skin, TEG and PV would be preferable [88,148,149]. However, the sources of energy from the human body generally occur in low-frequency or low-grade and in random form compared to the other applications. Designing efficient strategies to provide enough energy for monitoring and sensing in a limited environment is more challenging. Hybrid systems that use a variety of energy sources or conversion mechanisms could offer a way to address this challenge. Due to the ability to produce a huge amount of data that is important for healthcare, the impact of PE-TE hybrid nanogenerators (HBNGs) has recently come under study. This PE-TE HBNG measures the changes and diverse movements in the human body, including respiration, muscular contractions, and blood circulation. They can be used in various healthcare settings to power non-invasive sensors, enabling continuous patient monitoring without limiting the patient's comfort

or range of motion. Figure 10b schematically depicts various HBNG classes as well as numerous physiological conditions that these prospective dual-effect sensors can be used to monitor [150]. A wearable medical self-powered sensor system was designed for long-term healthcare applications by Mohsen et al. [151], as shown in Figure 10c. This system monitors the various parameters such as heartbeat, temperature, human body acceleration, and blood oxygen saturation in real time. It consists of sensors for pulse oximetry, temperature, and acceleration, a microcontroller unit, and a Bluetooth low-energy module. This sensor system typically relies on batteries for power, which have limited lifetimes. To overcome this limitation, a photovoltaic–thermoelectric hybrid energy harvester has been devised to provide continuous power to the wearable medical sensor system. The hybrid energy harvester incorporates a flexible photovoltaic panel, a thermoelectric generator module, a DC–DC boost converter, and two supercapacitors. Experimental results show that in active-sleep mode, the sensor system consumes an average power of 12.3 mW over 1 h, operating without the energy harvester for up to 46 h. These findings underscore the medical sensor system's sustainable and prolonged monitoring capabilities.

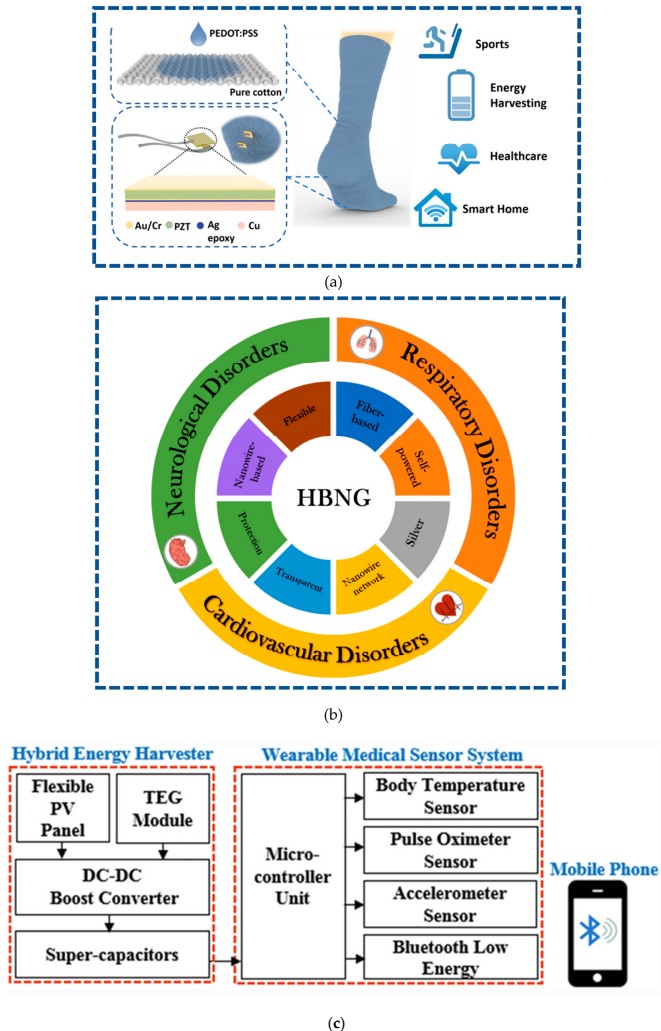

**Figure 10.** (**a**) Schematic diagram of hybrid PE-TE self-functional socks created by Zhu et al. [144]; (**b**) graphical overview of the different physiological problems that may be monitored using hybrid nanogenerators [150]; (**c**) schematic diagram of the self-powered hybrid system for healthcare applications by Mohsen et al. [151].

### 3.5. Aerospace Engineering

To improve passenger safety, decrease operational downtime, and save maintenance costs, it is essential to continuously monitor the operational conditions and structural integrity of spacecraft and aircraft [152,153]. For those distributed WSNs monitoring the operating condition, energy harvesting could offer reliable energy sources. Many energy sources are also available in spacecraft and aircraft, particularly vibration, temperature difference, and solar. A thorough analysis of the technology and energy sources for energy harvesting in aerospace applications is provided by Le et al. with a focus on thermal and vibration sources [154]. The main cause of interior vibration is the propulsion system. In both the propeller-driven aircraft and the helicopters, the amplitude and frequency range are greatly affected by the blade passage frequency and rotor speed. Common jet engine vibrations seem to have a frequency range of about 20–500 Hz. Engines, gear trains, and hydraulic systems all produce heat when used as a thermal source. Another way sensors located on the fuselage might be energized is by the temperature gradient between the fuselage and the cabin. Using the temperature gradient in the fuselage, Kiziroglou et al. developed a temperature gradient for a TEG utilizing the heat mass [155]. Wang et al. have created a self-powered jet engine monitoring system utilizing a non-linear PEH [156]. A 22 g energy harvester with a load of 100 kΩ may provide an output power of 79 mW at the rotational conditions' of 2050 rpm. In addition to the sources of energy on spacecraft, solar, diurnal temperature variations, and electromagnetic field in space can also be potential sources of energy. Because of the conditions in these kinds of environments, energy harvesting within this region is limited. Yet hybrid energy harvesting may be a future enabling technology for distributed and autonomous sensing in aerospace engineering. By using hybrid energy harvesters having multiple transduction mechanisms, it enhances the long-term monitoring of structural and environmental conditions and space exploration.

### 3.6. Industry Condition Monitoring

It is widely acknowledged that condition monitoring is essential for modern manufacturing and production processes particularly for smart industries (Industry 4.0; fourth technological revolution in the future). Since companies and industries cannot afford any unplanned downtime due to equipment failure; vibration, voltage, temperature, current, and other machine data are all fed to condition monitoring systems, which enables the early detection and evaluation of machine and system faults in real time. In addition, these equipment condition insights enhance productivity, expediting the transformation toward Industry 4.0. WSNs are suitable for implementing real-time condition monitoring because of their low power requirements and high flexibility. To supply power to sensor nodes, hybrid energy-harvesting systems gather unused energy through machines or the surrounding environment, minimizing the high cost and process of recharging or changing the batteries, particularly in remote or unreachable conditions [157]. A hybridized PE-EM-TE energy-harvesting system having a high output power was described by He et al. [82], as shown in Figure 11. One triboelectric nanogenerator (TENG), two piezoelectric generators (PEGs), and two electromagnetic generators (EMGs) make up the hybrid energy-harvesting system. In the EMG part, magnetic attractive forces cause a levitated annular magnet to oscillate vertically, absorbing vibration energy. PZT ceramic sheets are used in the PEG part. When the magnet's motion grasps them, they produce an electric current via the piezoelectric effect. When the magnet moves, the silicone and carbon nanotubes that are used in the TENG unit transmit triboelectric charges between the top and bottom layers, creating an electric current. Due to the movement of the levitated magnet, these three parts work together to generate energy by electromagnetic induction, the piezoelectric effect, and the triboelectric effect. To monitor the condition of the bearing and prevent mechanical damage or overload, the device was incorporated in a WSN of vibration and temperature. Future applications of industry IoTs will be highly dependent on it. An EM-PE-TEG HEH from industrial power equipment was presented by Yang et al. [126]. It

provides a constant DC voltage output that was applied to the Zig-Bee sensor to maintain its operation continuously.

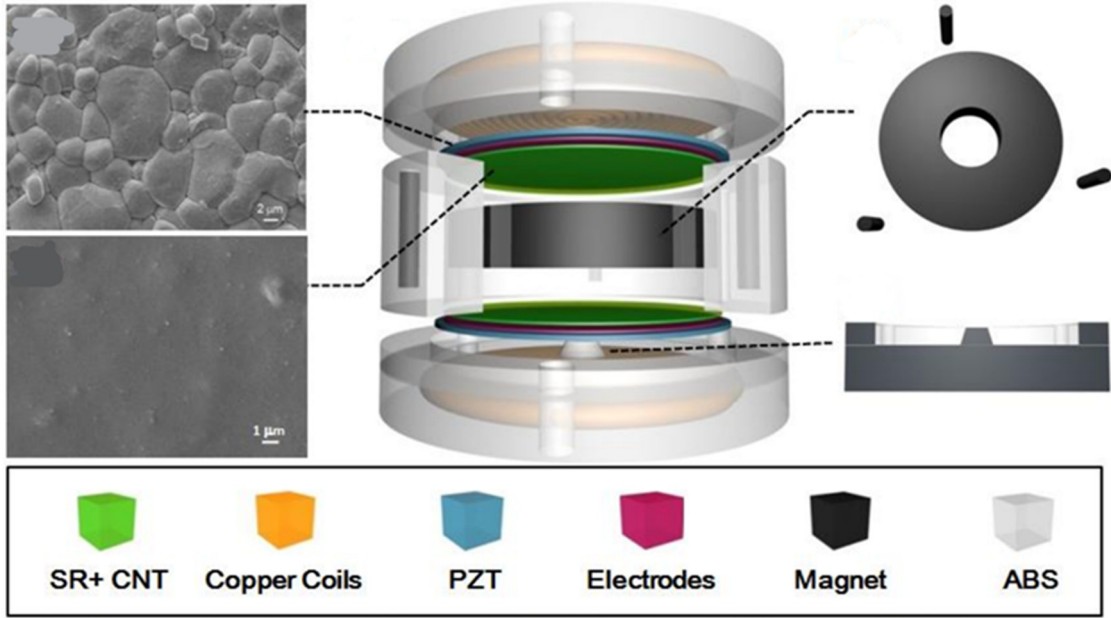

**Figure 11.** Hybridized PE-EM-TE energy-harvesting system developed by He et al. [82].

### 3.7. Water Purification

The potential of hybrid energy-harvesting systems to address the issue of obtaining reliable sources of energy for water purification has also attracted attention. Researchers are investigating how to sustain point-of-use (POU) water treatment technology using various energy sources like solar, thermal, and mechanical energy. Hybrid energy harvesters provide a constant and sustainable energy supply for water purification by integrating various energy conversion approaches. Various hybrid energy harvesters have been investigated for the water purification system: these are solar and triboelectric, solar, and piezoelectric, thermal-induced triboelectric, thermal-assisted piezoelectric, and thermal and photovoltaic hybrid energy harvesters. These hybrid energy harvesters have the potential to effectively harvest many energy sources simultaneously, making them reliable, affordable, and energy-efficient water-purifying solutions. They offer self-powered disinfection of microorganisms and degradation of pollutants, facilitating the purity of drinking water in areas with minimal access to electricity and sanitary facilities. Hybrid energy harvesters that are used for water purification become even more crucial during time of worldwide epidemics like COVID-19. To ensure continuous operation, hybrid energy harvesters, which combine solar and other energy sources, can efficiently harvest solar energy during the day and convert it to piezoelectric or triboelectric energy harvesting in changeable weather conditions or at night. The use of hybrid energy-harvesting devices for water purification presents a promising option for self-sufficient, sustainable water treatment. These systems help provide clean drinking water in various situations, utilizing the advantages of various energy sources. Figure 12 shows a schematic of a hybrid solar and thermal energy harvester, where water absorbs far-IR (infrared), photocatalytic layers absorb UV (ultraviolet), and photovoltaic solar cells absorb visible and near-IR [47,158].

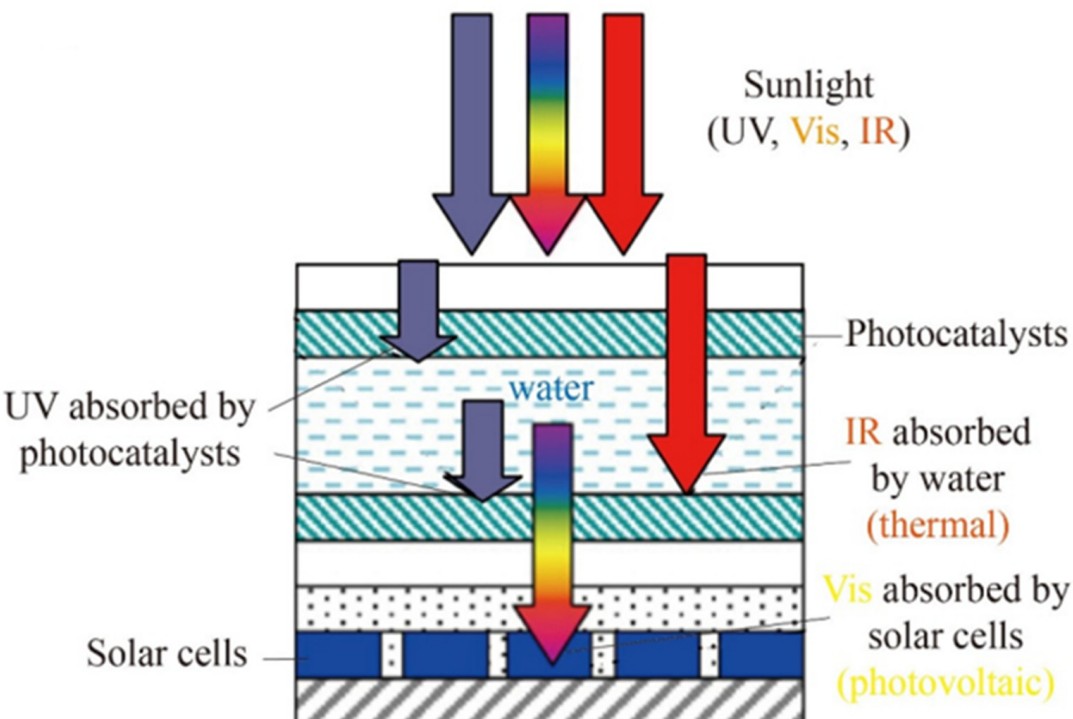

**Figure 12.** Schematic diagram of hybrid solar and thermal energy harvester for the water purification system [158].

Table 2 shows the hybrid energy systems that are utilized in the applications above. Yet, there is limited information, and few examples are available for the hybrid systems for the above applications. The area of hybrid energy harvesting is still in its infancy, and more research is needed to fully explore its potential. But ongoing work on hybrid energy harvesting shows that it has attracted a lot of attention from researchers and is included in one of the hot topics of today.

**Table 2.** Output performance and configurations of the hybrid energy systems that are utilized in the applications described above.

| Hybrid Energy-Harvesting System | Configuration | Output Performance | Advantages | Disadvantages | References |
|---|---|---|---|---|---|
| **Infrastructure Health Monitoring** | | | | | |
| Triboelectric–electromagnetic and solar cell | Integrated with WSN technology | Capable of lighting 100 of LEDs | Utilizes both mechanical and solar energy | Scalability is restricted due to reliance on WSN technology | [115] |
| Thermoelectric–electromagnetic generator | The turbine fan with magnets attached to the blades and is placed within the pipe | Thermoelectric power output 0.435 mW electromagnetic power output 0.584 mW | Dual thermoelectric and electromagnetic power generating | Very little power output | [132] |
| **Marine Monitoring and Development** | | | | | |
| Hybridized triboelectric–electromagnetic water wave energy harvester (WWEH) | Contain freely rotating magnetic sphere that detects the water's motion | Supercapacitor is charged by it to 1.84 V in 162 s | Makes use of a magnetic sphere that is free to rotate to detect water movements | Complex implementation and design | [137] |
| Triboelectric–electromagnetic hybridized nanogenerator | Chaotic pendulum | The maximum output power of triboelectric can reach 15.21 µW and the electromagnetic can reach 1.23 mW | High electromagnetic and triboelectric output power | A small scalability | [138] |
| Ship-shaped hybridized triboelectric–electromagnetic nanogenerator | Ship shaped have rolling magnetic cylinder in it | Peak power of 800 µW with an operating frequency of 2 Hz | A novel ship-shaped design with useable applications | Power output restrictions for some applications | [139] |

**Table 2.** *Cont.*

| Hybrid Energy-Harvesting System | Configuration | Output Performance | Advantages | Disadvantages | References |
|---|---|---|---|---|---|
| **Human Healthcare Monitoring** | | | | | |
| Hybrid piezoelectric–triboelectric self-functional socks | Self-functional socks of poly(3,4-ethylenedioxythiophene) polystyrenesulfonate | With a frequency of 2 Hz and a load resistance of 59.7 MΩ, an output power of 1.71 mW is obtained | Self-contained socks that continuously harvest energy by utilizing human motion | A low power output for medical equipment that is required | [144] |
| Photovoltaic–thermoelectric hybrid energy harvester | Consists of sensors for pulse oximetry, temperature, and acceleration, a microcontroller unit, and a Bluetooth low-energy module | In active sleep mode, the sensor system consumes an average power of 2.13 mW over 1 h, while it can operate without the energy harvester for up to 46 h | Extended battery life, sustainability through energy harvesting, and reliable long-term health parameter monitoring | Proper placement of photovoltaic panel, limitation in low light or low-temperature environment | [151] |
| **Industry Condition Monitoring** | | | | | |
| A hybridized piezoelectric–electromagnetic–triboelectric energy harvester | A single device with a magnetic levitation structure at its core and three harvest modes is incorporated | Under the frequency of 20 Hz, the output peak powers produce are TENG: 78.4 μW, EMG1: 36 mW, EMG2: 38.4 mW, PEG1: 122 mW, and PEG2: 105 mW. | Incorporates a variety of energy-harvesting techniques | Integrated and controlled complex device systems | [82] |
| Magnetic, thermoelectric, and vibration energy-harvesting system | - | Power produced by magnetic is 366 mW, thermoelectric is 1.98 W, and vibration is 0.63 mW | Uses a variety of energy sources to increase efficiency | Requires exact component alignment and location | [126] |
| **Water Purification** | | | | | |
| Photo-induced piezoelectric | ZnO nanowire | 92%, 10 min | Photo-induced piezoelectric water treatment that uses less energy | Specific uses for water treatment | [47] |
| TENG-assisted photocatalytic | Rotational TENG and visible light | 510 V 26 W/m | Combines photocatalysis and rotational TENG for energy harvesting | Only applicable to some photocatalytic applications | [47] |
| Thermal-induced piezoelectric | NaNbO₃ nanofibers | 86.5%, 80 min | Uses the thermally generated piezoelectric action to produce energy | A small scalability | [47] |
| Thermal-induced photocatalytic | UV: photocatalytic, far-IR: water, visible, and near-IR: solar cell | 300 W/m | Combines many energy sources to effectively purify water | Specific to uses for water treatment | [47] |

## 4. Challenges of Hybrid Energy Harvesters

The effective utilization of different energy sources within a single device offers a chance to sustain the power provision durability. Researchers have diligently explored this approach across various applications and energy sources. However, the integration of a hybrid energy-harvesting system presents different challenges as well that need to be addressed. The foremost challenge revolves around optimizing the synergy among various conversion mechanisms to enhance the overall system efficiency while maintaining a compact design. Additionally, the creation of power management circuits capable of efficiently handling the varied types of generated power is crucial.

As the research on the hybrid energy harvester is still early, conducting quantitative analyses or directly comparing various solutions proves challenging. PE-EM hybrid energy harvesters are primarily designed for vibration energy harvesting. However, there are some instances where they are utilized for harvesting human motion [159,160], airflow [161,162], and acoustic energy sources. On the other hand, PE-TE hybrid systems involving applying external forces to deformable laminated structures, achieving deformation of the piezoelectric material (e.g., PVDF) and contact-separation of the triboelectric materials (e.g., Al, Au, Cu, PDMS, PTFE) [163]. The piezoelectric part generally yields higher output power than the triboelectric part with the same dimensional area, while the triboelectric part primarily contributes to higher output voltage. Some studies have achieved the PE-TE dual effect through composite material synthesis like PDMS mixed with piezoelectric nanoparticles or nanofibers [164]. In triple-hybrid energy-harvesting systems, the output power is typically mainly contributed by one or two energy conversion effects. This suggests that certain energy conversion effects may not provide sufficient energy to the hybrid device but could

enhance the capacitor charging efficiency or act as a self-powered sensing unit. Attaining significant energy contribution from three types of energy conversion effects in one hybrid energy-harvesting system remains challenging.

The literature also showcases the integration of different conversion mechanisms, such as photovoltaic, thermoelectric, and piezoelectric, to harness various energy sources [165–168]. While hybrid systems offer more reliable power sources, many studies combine different functional materials with limited synergy. One distinctive concept highlighted in the literature is the use of PZT ceramics to harness both vibration and thermal energies through piezoelectric and pyroelectric effects [169] or the utilization of a piezoelectric beam with a magneto strictive mass to harness vibration and magnetic energy [170]. Power management circuits are the crucial yet demanding aspect of hybrid energy harvesting. Compared to mechanical design and material synthesis, the research focused on designing appropriate circuits for diverse hybrid systems is relatively underdeveloped. Presently, there are no universal solutions available for various hybrid energy harvesters. Still, some common architectures or mechanisms have been demonstrated, such as inductor/converter [171], maximum power point tracking [172] or synchronous control [173]. Enhancements in conversion efficiency, reduced electronics component count, and minimized power losses can all be achieved through a holistic design approach in power management circuits. With the future advancement of power management integrated circuits (PMICs) for hybrid energy harvesters, sustainable hybrid energy-harvesting systems are poisoned to become pivotal renewable energy technologies [174].

## 5. Future Perspective and Conclusions

In conclusion, this review on sustainable hybrid energy harvesting has shown the advancements made in this area and possible implications for the development of future sustainable energy solutions. Hybrid energy harvesting can be pivotal in enhancing energy efficiency and providing a green future. Researchers have shown that there has been an increase in the production of energy in the hybrid energy-harvesting systems through the integration of numerous energy conversion processes by assessing different materials, configurations, and strategies, thereby increasing the output power and overall efficiency. The area of hybrid energy harvesting is still in its infancy, and more study is needed to fully explore its potential. The ability of hybrid energy harvesting to combine several energy conversion processes enables it to continuously generate energy even in various environmental conditions and is one of its notable benefits. They combine several energy conversion processes, including triboelectric, piezoelectric, and electromagnetic, allowing the utilization of a variety of energy sources, including vibrations, pressing force, rotating motion, and wind flow. This adaptability opens new opportunities for generating sustainable power in various applications, like smart transportation, infrastructure health monitoring, marine monitoring and development, healthcare monitoring, aerospace engineering, industry condition monitoring, and water purification. However, as the acquired results are thoroughly examined, it is discovered that it is difficult to compare and assess various hybrid energy-harvesting systems thoroughly due to the lack of varying input quantities and standardized specification specifications. Future studies should aim to create a unified specification standard and carry out rigorous performance comparisons to determine the most reliable and effective hybrid energy harvesting techniques. Hybrid energy-harvesting systems hold significant promise for addressing the rising need for sustainable energy solutions in the upcoming years by filling these gaps and improving energy conversion efficiency and power density.

By comparing the performance comparison of the several hybrid energy harvesters mentioned in Table 1, several noteworthy results are obtained. Khan et al. [90] talked about a piezoelectric–electromagnetic hybrid energy harvester with an output performance of 49 μW for piezoelectric and 3.2 μW for electromagnetic. Similarly, Coa et al. [68] used an electromagnetic–triboelectric hybrid energy harvester utilizing a coil/magnet in horizontal sliding mode and Cu/FEP in a lateral sliding mode, achieving an output of

13.8 $\mu$W/cm$^3$. The various configurations and performance levels possible with the hybrid energy harvesters can be seen in these examples. Hybrid systems can improve power generation and offer more versatile and reliable energy-harvesting options by combining multiple energy conversion techniques. Selecting the best hybrid energy harvester without further context and precise criteria is difficult. The selection of the suitable hybrid energy harvester system depends upon various factors, i.e., application, source of energy, output performance, efficiency, cost-effectiveness, and device size, all of which affect the system's performance. However, the highest output performance shown in Table 1 is for the electromagnetic–triboelectric hybrid energy harvester having a coil/magnet in approaching separation mode and Al/PDMS in lateral sliding mode with an output performance of 381 $\mu$W/cm$^3$. This system can be used in applications where human motion and vibration are present.

To enhance hybrid energy harvesting, it is recommended to improve energy conversion efficiencies, develop advanced integration techniques, and investigate new hybrid configurations. By improving energy conversion efficiency, more power could be produced from the energy source that is accessible, improving performance and expanding the capacity for energy harvesting. A single system may more effectively integrate several energy-harvesting processes, including thermoelectric, electromagnetic, and piezoelectric due to enhanced integration techniques. This integration can increase the overall power production by effectively utilizing several energy sources simultaneously. Additionally, investigating novel hybrid configurations may reveal innovative and effective energy-harvesting techniques. By focusing on these, researchers can enhance and develop the field of hybrid energy harvesting, realizing its full potential for the sustainable powering of various systems and devices.

**Author Contributions:** H.S.: data curation, software, writing—original draft; A.A.: data curation, software, writing—original draft; S.A.: data curation, software, writing—original draft; A.A.: formal analysis, review, and editing; S.A.: resources, data curation, review, and editing; S.A.: formal analysis, review, and editing; W.A.A.: resources, supervision, review and editing; M.N.: resources, supervision, review and editing; S.A.K. and H.S.: resources, supervision, review and editing. All authors have read and agreed to the published version of the manuscript.

**Funding:** This research received no external funding.

**Data Availability Statement:** Data are contained within the article.

**Conflicts of Interest:** The authors declare no conflict of interest.

## Nomenclature

| | |
|---|---|
| Wireless Sensor | WSN |
| Hybrid Energy Harvester | HEHs |
| Piezoelectric–Electromagnetic | PE-EM |
| Triboelectric Nanogenerator | TENG |
| Piezoelectric–Triboelectric | PE-TE |
| Barium Titanate | BTO |
| Polydimethylsiloxane | PDMS |
| Polyvinylidene Fluoride | PVDF |
| Electromagnetic Energy Harvester | EMEHs |
| Polytetrafluoroethylene | PTFE |
| Piezoelectric–Electromagnetic–Triboelectric | PE-EM-TE |
| Piezoelectric Energy Harvester | PEH |
| Electromagnetic Generators | EMGs |

| Water-Wave Energy Harvester | WWEH |
| Ship-Shaped Hybridized Nanogenerator | SHNG |
| Hybrid Nanogenerator | HBNGs |
| Piezoelectric Generators | PEG |
| Lead Zirconate Titanate | PZT |
| Point-Of-Use | POU |
| Infrared | IR |
| Ultraviolet | UV |
| Power Management Integrated Circuits | PMICs |

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
