# Peer review of "Applications of Sustainable Hybrid Energy Harvesting: A Review"

_jlpea, doi:10.3390/jlpea13040062_

Round 1

Reviewer 1 Report

Comments and Suggestions for Authors

Applications of Sustainable Hybrid Energy Harvesting: A Review

This review is rich in content, has conducted extensive research and summary, and needs to be strengthened in terms of writing logic and classification by major revision.

1. It is best to provide frequency domain range (e.g:10-70 Hz) rather than size (60 Hz) ‘Xia et al. created a novel approach for the piezoelectric-electromagnetic (PE-EM) harvesters [60]. The results showed a broad operating range of about 62 Hz.

2. The writing logic needs to be strengthened, and the proposed viewpoints need to be explained in conjunction with the literature. For example, there is no further text on the circuit after this sentence: To make the PE-EM HEHs more advantageous, it is essential to find more efficient ways to control two power sources.

3. Some statements need to be considered comprehensively. There are two distinct methods for converting mechanical energy into electrical energy: the piezoelectric effect and the triboelectric effect. Please change as: Besides electromagnetic generator is based on electromagnetic induction, the piezoelectric effect and the triboelectric effect are based on displacement current for converting mechanical energy into electrical energy. 

4. Please abbreviate in the table, such as Piezoelectric-electromagnetic as PE-EM; Piezoelectric-triboelectric as PE-TENG;

5. The chapter 3 has section 3.1, 3.2...Why not 2.1, 2.2 has section names? It is recommended to distinguish by the hierarchy of hybrid power generation materials, hybrid power generation structures, and hybrid power generation circuits.

6. In chapter 3, firstly, it is necessary to describe the characteristics of energy, such as light intensity, wind energy size, vibration frequency and amplitude. Furthermore, how to collect energy, what key technologies are used, what effects are obtained, and what specific applications are used.

7. Summary of Table 2 is required in chapter 3 for Applications of Sustainable Hybrid Energy Harvesting like Table 1.

8. The work of hybrid energy harvesting for self powering and self sensing was not mentioned. Such as:

https://doi.org/10.1016/j.nanoen.2020.105555

9. In Future Perspective and Conclusion, Many literature mentions the specification standard, power management integrated circuits, but the author did not cite or analyze them. It is suggested to provide a more specific perspective.

10. The conclusion in the last paragraph is confusing, and it is necessary to reorganize the language to express the prospects of combining photovoltaic and other all-weather energy harvesting.

Comments on the Quality of English Language

ok

Author Response

This review is rich in content, has conducted extensive research and summary, and needs to be strengthened in terms of writing logic and classification by major revision.

[Comment]: 1. It is best to provide frequency domain range (e.g:10-70 Hz) rather than size (60 Hz) ‘Xia et al. created a novel approach for the piezoelectric-electromagnetic (PE-EM) harvesters [60]. The results showed a broad operating range of about 62 Hz.’

[Response]: Thank you for your valuable suggestion regarding the clarification of the operating frequency range. We have carefully addressed this concern and revised the text accordingly. In line 198-199 of the article, we have amended the statement to clearly indicate the frequency domain range, as recommended. The updated sentence now reads, “Xia et al. created a novel approach for the piezoelectric-electromagnetic (PE-EM) harvesters [60]. The results showed a broad operating frequency range of about 25.5-62 Hz”.

[Comment]: 2. The writing logic needs to be strengthened, and the proposed viewpoints need to be explained in conjunction with the literature. For example, there is no further text on the circuit after this sentence: ‘To make the PE-EM HEHs more advantageous, it is essential to find more efficient ways to control two power sources.’

[Response]: We appreciate your feedback. In response to this valuable suggestion, we have revised this section. We have extended the discussion from line 201-220 and provided a more detailed explanation. This added content, along with the inclusion of Figure 2(c) (which is a block circuit diagram), serves to clarify the point and enhance the overall clarity of the paper.

[Comment]: 3. Some statements need to be considered comprehensively. ‘There are two distinct methods for converting mechanical energy into electrical energy: the piezoelectric effect and the triboelectric effect.’ Please change as: ‘Besides electromagnetic generator is based on electromagnetic induction; the piezoelectric effect and the triboelectric effect are based on displacement current for converting mechanical energy into electrical energy.’ 

[Response]: We have carefully considered your suggestion and made the amendment from line 250-260. The revised statements now provides a more comprehensive perspective on the methods for converting mechanical energy into electrical energy.

[Comment]: 4. Please abbreviate in the table, such as ‘Piezoelectric-electromagnetic’ as ‘PE-EM’; ‘Piezoelectric-triboelectric’ as ‘PE-TENG’

[Response]: As per your suggestion, we have updated the table to include abbreviations. Furthermore, to provide a comprehensive overview of the abbreviations used in the paper, we have included a nomenclature table to enhance clarity for the readers.

[Comment]: 5. The chapter 3 has section 3.1, 3.2...Why not 2.1, 2.2 has section names? It is recommended to distinguish by the hierarchy of hybrid power generation materials, hybrid power generation structures, and hybrid power generation circuits.

[Response]: Thank you for your comment. We've taken your suggestion into consideration and added subsections with appropriate headings to Chapter 2, differentiating them as 2.1, 2.2, and so forth.

[Comment]: 6. In chapter 3, firstly, it is necessary to describe the characteristics of energy, such as light intensity, wind energy size, vibration frequency and amplitude. Furthermore, how to collect energy, what key technologies are used, what effects are obtained, and what specific applications are used.

[Response]: In response to your valuable feedback, we have carefully revised Chapter (Section) 3 of our article. This section now provides a detailed description of the characteristics of various energy sources, explained the methods employed to collect energy, including the key technologies used and discussed the effects obtained from these energy sources and have highlighted specific applications. To enhance the clarity, we have integrated a general introduction in the initial paragraph of this section to provide a comprehensive overview. To better illustrate the output effects of these technologies, we incorporated Table 2 into this section.

[Comment]: 7. Summary of Table 2 is required in chapter 3 for Applications of Sustainable Hybrid Energy Harvesting like Table 1.

[Response]: As per suggestions Table 2 have been added in order to address the above comment. This table provides an overview of the output performance and configurations of the hybrid energy systems, enhancing the comprehensiveness of our discussion on the applications of sustainable hybrid energy harvesting.

[Comment]: 8. The work of hybrid energy harvesting for self-powering and self-sensing was not mentioned. Such as:

https://doi.org/10.1016/j.nanoen.2020.105555

[Response]: We appreciate your suggestion. In the Introduction section, we have expanded our discussion for hybrid energy harvesting as self-powered and self-sensing, include reference to such work, from lines 72-81. Additionally, in Section 3, there are few examples of applications of hybrid energy harvesting that serve as self-powered and self-sensing systems. For instance, we have cited the work done by Mohsen et al. from line 581-594 on a wearable medical self-powered sensor system. These amendments enhance the comprehensiveness of our paper by addressing the specific topic you highlighted.

[Comment]: 9. In Future Perspective and Conclusion, Many literature mentions the specification standard, power management integrated circuits, but the author did not cite or analyze them. It is suggested to provide a more specific perspective.

[Response]:  Thank you for your feedback regarding the Future Perspective and Conclusion of the paper. Considering your feedback, we have revised this whole section and rewritten it to provide a concise summary of the main outcomes and significant findings presented in the paper, along with some recommendations. Furthermore, we have added a new section Challenges of Hybrid Energy Harvesters, providing a more specific perspective. This addition enhances the comprehensiveness of our paper by discussing the challenges faced in the field of hybrid energy harvesting and offers insights into potential areas of future research. Your input has significantly contributed to improving the quality of our paper.

[Comment]: 10. The conclusion in the last paragraph is confusing, and it is necessary to reorganize the language to express the prospects of combining photovoltaic and other all-weather energy harvesting.

[Response]: We apologize for any confusion caused and appreciate the opportunity to address your concerns. We have rewritten the Future Perspective and Conclusion section again to enhance the clarity of our paper.

Reviewer 2 Report

Comments and Suggestions for Authors

This paper reviews different sustainable hybrid energy harvesting methods to achieve more efficient and continuous energy supply. At the same time, the application scenarios of different energy harvesters have also been further discussed. However, some minor revisions are required for publication.

1.      When introducing hybrid energy collection methods, pictures from relevant articles should be quoted to illustrate the content, such as device structure schematics or working principal schematics.

2.      Why not continue to discuss the shortcomings of current hybrid energy harvesting and its possible improvements? For example, the shortcomings when used in human energy harvesting and feasible solutions.

3.      Table 1 lists the comparative data of hybrid energy harvesters, which should be simply classified and summarized according to their features. For example, harvesters that are suitable for human body energy harvesting (miniaturization, portability, comfort, etc.).

4.      The application methods and scenarios of various hybrid energy harvesters in different fields should be described more specifically. For example, to harvest human energy for health monitoring, the application scenarios of related devices should be described in more details.

5.      The specific working method of the hybrid energy harvester is not mentioned in this paper. Are they a simple combination of different harvesters? Will they produce coupling effects during operation that enhance the output? May they have a certain impact on each other and reduce the output? The author should give some introduction in the second part ‘Hybrid Energy Harvesting’.

6.      Some important references (J. Low Power Electron. Appl. 2023, 13, 11; Biosens. Bioelectron. 2021, 187, 113329; Adv. Funct. Mater. 2023, 33, 2213410) are highly related to this work and should be cited appropriately.

Comments on the Quality of English Language

Minor editing of English language is required.

Author Response

This paper reviews different sustainable hybrid energy harvesting methods to achieve more efficient and continuous energy supply. At the same time, the application scenarios of different energy harvesters have also been further discussed. However, some minor revisions are required for publication.

[Comment]: 1. When introducing hybrid energy collection methods, pictures from relevant articles should be quoted to illustrate the content, such as device structure schematics or working principal schematics.

[Response]: We appreciate your valuable suggestion. In response to your feedback, we have carefully revised the Section 2. We have incorporated figures and content to enhance the clarity of our paper. These additions include device structure schematics and working principle schematics from relevant articles, as recommended. By including these visuals, we aim to better illustrate the content and provide a more comprehensive understanding of the discussed methods.

[Comment]: 2. Why not continue to discuss the shortcomings of current hybrid energy harvesting and its possible improvements? For example, the shortcomings when used in human energy harvesting and feasible solutions.

[Response]:  Thank you for your valuable comment. We have made significant changes to address your suggestion. In the introduction section, we have added additional paragraphs (line 116-158) that highlight some of the limitations and challenges associated with hybrid energy harvesting.

Furthermore, Section 2 contains paragraphs that elaborates on the drawbacks of various hybrid energy harvesting methods, with some detailed information presented from line 382-392.

In addition, we have introduced a new Section 4 in our paper dedicated to discussing the Challenges of Hybrid Energy Harvesting. This section provides a comprehensive overview of the shortcomings and potential areas of improvement. In Section 5, "Future Recommendations and Conclusion," we have research recommendations that progress and promote the use of this technology.

[Comment]: 3. Table 1 lists the comparative data of hybrid energy harvesters, which should be simply classified and summarized according to their features. For example, harvesters that are suitable for human body energy harvesting (miniaturization, portability, comfort, etc.).

[Response]: We appreciate your input on reorganizing Table 1 for improved clarity and understanding. In response to your suggestion, we believe that presenting the hybrid energy harvesters based on the energy transduction mechanisms involved can indeed enhance the table's readability and overall utility. This approach will categorize the harvesters based on the methods involved in energy transduction, making it easier for readers to identify and compare the different types of hybrid energy harvesting systems.  We would like to express our gratitude for this valuable suggestion, which we believe will enhance the quality of our paper. Your feedback is invaluable to us, and we are committed to delivering an improved manuscript based on your input.

[Comment]: 4. The application methods and scenarios of various hybrid energy harvesters in different fields should be described more specifically. For example, to harvest human energy for health monitoring, the application scenarios of related devices should be described in more details.

[Response]: In response to this valuable feedback, we have revised the Section 3 again. We have expanded the descriptions of the application methods and scenarios for each type of hybrid energy harvester, particularly we have added the work of Mohsen et al., which is showing the self-powered sensor system for health monitoring (Line 581-594). To further improve the clarity and readability of the section, we have introduced a new Table 2. This table summarizes the output performance and configuration details of different hybrid energy harvesters based on their applications, as described in this Section 3. We have restructured the section to present the application scenarios in a more organized and coherent manner. We believe these revisions will significantly enhance the readability, clarity, and practical value of the paper.

[Comment]: 5. The specific working method of the hybrid energy harvester is not mentioned in this paper. Are they a simple combination of different harvesters? Will they produce coupling effects during operation that enhance the output? May they have a certain impact on each other and reduce the output? The author should give some introduction in the second part ‘Hybrid Energy Harvesting’.

[Response]: Thank you for your insightful comment regarding the explanation of hybrid energy harvesters in the Section 2. We have thoroughly revised this section, incorporating additional details from line 176-184 as suggested to improve the clarity.

[Comment]: 6. Some important references (J. Low Power Electron. Appl. 2023, 13, 11; Biosens. Bioelectron. 2021, 187, 113329; Adv. Funct. Mater. 2023, 33, 2213410) are highly related to this work and should be cited appropriately

[Response]: These references are added in the manuscript in order to address the above comment .

Reviewer 3 Report

Comments and Suggestions for Authors

This manuscript reports a review on sustainable hybrid energy harvesting, including some applications. This review includes the output performance of several hybrid energy harvesters. However, this review can be improved based on the following issues:

1.- This manuscript should consider more recent references between 2020 and 2023 about green energy harvesters.

2.- The introduction section has a large paragraph. In addition, the introduction should consider the advantages and challenges of hybrid energy harvesters compared to non-hybrid energy harvesters.

3.- The second section can incorporate more discussions on the advantages and drawbacks of the different transduction mechanisms, such as piezoelectric-electromagnetic, piezoelectric-triboelectric, electromagnetic-triboelectric, piezoelectric-electromagnetic-triboelectric, piezoelectric and pyroelectric,  electromagnetic, thermoelectric, and piezoelectric. Furthermore, this section can include figures and descriptions of the working principles of the different hybrid energy harvesters. 

4.- The third section can consider more applications or future applications of hybrid energy harvesters, by considering different disciplines. This section could incorporate more discussions on the output performance and reliability of the hybrid energy harvesters.

5.- The resolution of all the figures must be enhanced.

6.- This review could add a section on the challenges of hybrid harvesters, regarding their design, fabrication processes, packaging, signal processing, and reliability. 

Comments on the Quality of English Language

The English grammar and style are acceptable.

Author Response

This manuscript reports a review on sustainable hybrid energy harvesting, including some applications. This review includes the output performance of several hybrid energy harvesters. However, this review can be improved based on the following issues:

[Comment]: 1.- This manuscript should consider more recent references between 2020 and 2023 about green energy harvesters.

[Response]: We appreciate your valuable input regarding the incorporation of more recent references. In response to this suggestion, we have added recent references to the manuscript. These new citations enhance the timeliness and relevance of the paper, ensuring that readers have access to the latest developments.

[Comment]: 2.- The introduction section has a large paragraph. In addition, the introduction should consider the advantages and challenges of hybrid energy harvesters compared to non-hybrid energy harvesters.

[Response]: Thank you for your suggestions. In response to this feedback, we have made significant revisions to the introduction section. Specifically, we have added an additional paragraphs that aims to provide a comprehensive overview of the advantages and challenges associated with hybrid energy harvesters. Moreover, we try to better explain the research and provide a critical literature review. This paragraphs highlight the trends, patterns, ideas, comparisons, and relationships among the studies to better identify the gaps and explore areas with high potential in hybrid energy harvesting.

[Comment]: 3.- The second section can incorporate more discussions on the advantages and drawbacks of the different transduction mechanisms, such as piezoelectric-electromagnetic, piezoelectric-triboelectric, electromagnetic-triboelectric, piezoelectric-electromagnetic-triboelectric, piezoelectric and pyroelectric,  electromagnetic, thermoelectric, and piezoelectric. Furthermore, this section can include figures and descriptions of the working principles of the different hybrid energy harvesters. 

[Response]:  Thank you for your insightful comment. To address it effectively, we have carefully revised this section 2. We have incorporated figures and content to enhance the clarity of our paper. These additions include device structure schematics and working principle schematics from relevant articles, as recommended. By including these visuals, we aim to better illustrate the content and provide a more comprehensive understanding of the discussed methods. Moreover, we have added subsections with appropriate headings to Chapter 2, differentiating them as 2.1, 2.2, and so forth. In the last paragraph of this section 2 from line 382-392 we have incorporated discussion on the advantages and disadvantages of different types of hybrid energy harvesters. Further challenges that are faced by the hybrid energy harvesters are mentioned in the section 4 of the manuscript.

[Comment]: 4.- The third section can consider more applications or future applications of hybrid energy harvesters, by considering different disciplines. This section could incorporate more discussions on the output performance and reliability of the hybrid energy harvesters.

[Response]: We appreciate reviewer’s input. Based on the above comments, we have revised this section 3. We have also added another heading “Water Purification” in the application section to further enhance the clarity and to show some more areas with high potential in hybrid energy harvesting.   We have expanded the descriptions of the application methods and scenarios for each type of hybrid energy harvester. To further improve the clarity and readability of the section, we have introduced a new Table 2. This table summarizes the output performance and configuration details of different hybrid energy harvesters based on their applications, as described in this Section 3. We have restructured the section to present the application scenarios in a more organized and coherent manner. We believe these revisions will significantly enhance the readability, clarity, and practical value of the paper.

[Comment]: 5.- The resolution of all the figures must be enhanced.

[Response]: We have increased the resolution of all figures as per your suggestion.

[Comment]: 6.- This review could add a section on the challenges of hybrid harvesters, regarding their design, fabrication processes, packaging, signal processing, and reliability. 

[Response]: Thank you for your valuable suggestion. We have incorporated your recommendation by adding a new Section 4 in the manuscript to comprehensively address the challenges associated with hybrid energy harvesters. Your input has significantly enhanced the completeness of our review.

Round 2

Reviewer 1 Report

Comments and Suggestions for Authors

The author has made significant revisions. I agree to accept it.

Comments on the Quality of English Language

it is ok

Reviewer 3 Report

Comments and Suggestions for Authors

The authors have addressed all the reviewer's comments.

Comments on the Quality of English Language

The English grammar is good.